# A licensing step links AID to transcription elongation for mutagenesis in B cells

Stephen P. Methot [1,2,9], Ludivine C. Litzler[1,3], Poorani Ganesh Subramani[1,2], Anil K. Eranki[1], Heather Fifield[4], Anne-Marie Patenaude[1,10], Julian C. Gilmore[1], Gabriel E. Santiago[5], Halil Bagci[1,6], Jean-François Côté[1,6,7], Mani Larijani[4], Ramiro E. Verdun[5,8] & Javier M. Di Noia [1,2,3,7]

Activation-induced deaminase (AID) mutates the immunoglobulin (*Ig*) genes to initiate somatic hypermutation (SHM) and class switch recombination (CSR) in B cells, thus underpinning antibody responses. AID mutates a few hundred other loci, but most AID-occupied genes are spared. The mechanisms underlying productive deamination versus non-productive AID targeting are unclear. Here we show that three clustered arginine residues define a functional AID domain required for SHM, CSR, and off-target activity in B cells without affecting AID deaminase activity or *Escherichia coli* mutagenesis. Both wt AID and mutants with single amino acid replacements in this domain broadly associate with Spt5 and chromatin and occupy the promoter of AID target genes. However, mutant AID fails to occupy the corresponding gene bodies and loses association with transcription elongation factors. Thus AID mutagenic activity is determined not by locus occupancy but by a licensing mechanism, which couples AID to transcription elongation.

[1] Institut de Recherches Cliniques de Montréal, 110 av.des Pins Ouest, Montréal, QC H2W 1R7, Canada. [2] Department of Medicine, Division of Experimental Medicine, McGill University, 1001 Boulevard Decarie, Montreal, QC H4A 3J1,, Canada. [3] Department of Biochemistry and Molecular Medicine, 2900 boul., Édouard-Montpetit, Montréal, QC H3T 1J4, Canada. [4] Department of BioMedical Sciences, Faculty of Medicine, Memorial University of Newfoundland, St. John's, NL A1B 3B6, Canada. [5] Department of Medicine, Division of Hematology, Sylvester Comprehensive Cancer Center, University of Miami, Miami, FL 33136, USA. [6] Department of Anatomy and Cell Biology, McGill University, 3640 rue University, Montréal, QC H3A 0C7, Canada. [7] Department of Medicine, Université de Montréal, C.P. 6128, succ. Centre-ville, Montréal, QC H3C 3J7, Canada. [8] Geriatric Research, Education and Clinical Center, Miami Veterans Affairs Healthcare System, Miami, FL 33125, USA. [9] Present address: Friedrich Miescher Institute for Biomedical Research, Maulbeerstrasse 66, R-1066.2.58, 4058 Basel, Switzerland. [10] Present address: Genos BioCentar Borongajska cesta 83H, 10000 Zagreb, Croatia. These authors contributed equally: Stephen P. Methot, Ludivine C. Litzler.  Correspondence and requests for materials should be addressed to J.M.D.N. (email: javier.di.noia@ircm.qc.ca)

The enzyme activation-induced deaminase (AICDA, referred to as AID, encoded by the *Aicda* gene) initiates genetic modifications at the immunoglobulin (*Ig*) genes in activated B cells[1,2]. AID catalyses the deamination of deoxycytidine to deoxyuridine on single-stranded DNA (ssDNA)[2]. This change is mutagenic, but further processing of the deoxyuridines by DNA repair enzymes underpins somatic hypermutation (SHM) and class switch recombination (CSR), which are indispensable for efficient antibody responses[1–3]. As deleterious side effects of SHM and CSR, AID can mutate and induce DNA damage outside the *Ig* loci, in many cases triggering chromosomal translocations[4].

DNA repair pathways limit off-target mutations and DNA damage by AID[5–7]. Nevertheless, several additional layers of regulation are necessary to control AID oncogenic and cytotoxic activity[8]. Regulation of AID protein levels and nuclear access restrains both on- and off-target activities, but it is unclear whether they contribute to target specificity[1]. The preferential targeting of AID to the *Ig* genes and how AID mutates a small number of additional genomic loci while sparing most others is an area of active research[4,9]. The *Ig* loci possess an intrinsic ability to attract AID activity[10], conferred in part by specialized *cis*-acting elements that combine transcriptional enhancers with multiple transcription factor-binding sites and can ectopically function to target SHM[11,12]. Similar elements have not been identified in AID off-targets, but these loci share with the *Ig* the characteristic of showing convergent transcription and being associated with strong super-enhancers[13–15]. Nonetheless, many highly transcribed genes have similar characteristics but are not mutated, so an additional layer of regulation must exist. The identity of the *trans*-acting factors targeting AID to the *Ig* loci is also elusive, though non-coding RNA and transcription factors likely have a function[4]. Genome-wide studies have identified a few factors that correlate with AID occupancy and mutagenic activity, such as RNA polymerase II (RNAPII), its associated factor Spt5 (Supt5h) and the RNA processing exosome[16–18]. Again, these factors function at a much larger number of loci than are mutated by AID and fail to explain AID's specificity on their own.

Thus there is a three-tier system of AID targeting, with the *Ig* loci being targeted much more frequently than any AID off-targets but the latter restricted to a few hundred sites. Beyond specific examples of loci occupied but not mutated by AID[19], the analysis of AID occupancy by chromatin immunoprecipitation (ChIP)–sequencing has suggested its association with ~6000 genes in B cells, while AID-induced damage is limited to some 300 loci[7,13,14,20,21]. This begs the question of why most sites bound by AID are spared from its activity.

Here we report a new functional domain of AID that is dispensable for enzymatic activity but necessary for on- and off-target biological activity in B cells. Systematic analysis of the function and interactome of AID variants with mutations in this arginine-rich (RR) domain reveals that they have a defect specifically in their association with the gene body of physiological and collateral target sites, explaining their failure to mutate. Our results uncover a licensing mechanism that most likely couples AID to transcription elongation, which can explain why occupancy is not sufficient to predict AID activity and suggest a new model for productive AID targeting. Our data also suggest that limiting nuclear levels of AID are important to enforce this licensing mechanism.

## Results

### Three arginines in AID α6 define a new functional domain

In previous structure–function analyses, we used a set of chimeric proteins in which contiguous regions of AID were replaced by their homologous region from APOBEC2 (A2)[22–24]. Only one of these, AID-A2#5, could mutate the *Escherichia coli* genome (Supplementary Fig. 1a, b). AID-A2#5 replaces a large C-terminal portion of AID, starting from the loop preceding alpha-helix 6 (α6) and eliminating the C-terminal E5 domain, which is necessary for CSR[25]. However, not only did adding back E5 not rescue CSR but this chimera also lacked IgV SHM activity when used to complement *Aicda*^−/− B cells (Supplementary Fig. 1c, d). A smaller chimera, replacing only the α6 of AID with that of A2 (AID-A2 α6) had measurable activity in *E. coli* but not in B cells (Supplementary Fig. 1a–d). The functional defect of AID-A2 α6 could not be explained by differences in protein abundance or nuclear access (Supplementary Fig. 1b–e). These results suggested that the AID α6 contained residues required for SHM and CSR but dispensable to mutate *E. coli*.

We sought to identify single amino acid substitutions within AID α6 that could separate its ability to mutate *E. coli* from its biological activity in B cells. Comparing a three-dimensional molecular model of AID[26] to the A2 structure[27] showed several residue and charge differences in α6 between these paralogues (Fig. 1a). To obtain AID variants with minimal structural alterations that could recapitulate the phenotype of the chimeras, we independently mutated several of these AID residues to the corresponding A2 residue. Three of these recapitulated the results obtained with the chimeras. AID R171Y, R174E and R178D mutated *E. coli* with the same efficiency as AID but were inactive for SHM and CSR (Fig. 1b–d). In contrast, adjacent mutations AID R177A and S173E maintained all three activities (Fig. 1a–d). Notably, Arg 171, 174 and 178 are conserved in AID from most jawed vertebrates but not in the APOBECs (Supplementary Fig. 1f) and form a contiguous AID surface (Fig. 1a)[26,28,29]. The natural AID variant R174S found in some immunodeficient HIGM2 patients[30] conserves DNA binding and processivity but has substantially reduced catalytic activity[31]. Furthermore, protein arginine residues are common contact points with nucleic acids[32]. Nonetheless, the R-mutants show similar DNA-binding affinity and deaminate ssDNA within a bubble substrate in vitro with similar specific activities to wild-type (wt) AID (Fig. 1e, f).

To test the role of the positive charge contributed by these Arg residues, we substituted each of them for lysine (Supplementary Fig. 2). AID R171K had reduced *E. coli* mutation activity and a proportional decrease in SHM and CSR, indicating a structural contribution to catalysis that Tyr can provide but Lys cannot. AID R174K mutated *E. coli* with 50% efficiency compared to AID and showed a proportional decrease in SHM but lacked CSR, indicating not only a structural contribution for R174 but also a specific role in CSR. AID R178K was indistinguishable from AID for all activities, indicating that, in this case, the charge is sufficient for biological function.

We conclude that AID arginines 171, 174 and 178 not only contribute in charge but also structurally to creating a functional domain necessary for SHM and CSR. Since the substitutions to the corresponding A2 residues consistently retained enzymatic activity, we used those mutants for dissecting the role of this functional domain (hereafter, the RR domain).

### AID R-mutants enter the nucleus but lack off-target activity.

Nuclear access of AID is restricted, with ~10% of AID being nuclear in homeostasis as a consequence of several mechanisms regulating AID nuclear residency and protein stability[33]. To exclude that the functional defect of the R-mutants was due to defective nuclear access, we analysed their subcellular localization. We used AID-deficient CH12 B cells, in which wt AID reconstituted CSR activity but none of the R-mutants did, despite similar expression levels (Supplementary Fig. 3a-c). Akin to wt

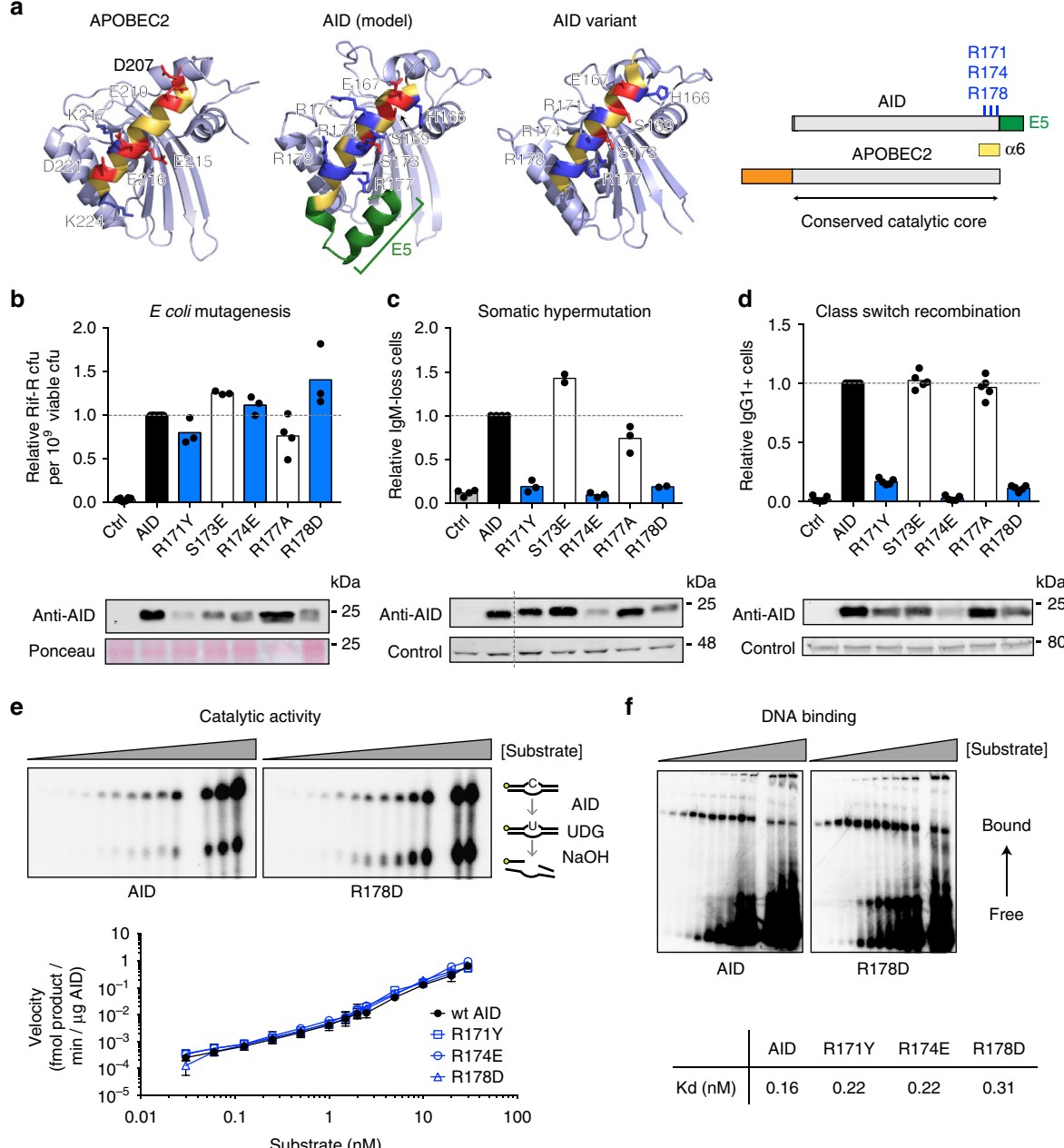

**Fig. 1** Identification of functionally inactive AID variants. **a** (Left) Comparison of our 3D model of AID to the experimental structure of APOBEC2 (PDB 2NYT) and an AID variant (PDB 5JJ4) in the same orientation. Selected residues within helix α6 are labelled. Side chains for basic (blue) and acidic (red) residues are shown. (Right) Schemes of AID and APOBEC2. **b** Mutagenic activity in *E. coli*, measured by the relative frequency of rifampicin-resistant (Rif-R) colony-forming units (cfu) arising from cultures expressing AID variants or empty vector (Ctrl). Means (bars) of median values (dots) obtained from 3 to 8 independent experiments (5 cultures/experiment) per construct are shown, normalized to AID. **c** Somatic hypermutation activity, assayed by the relative IgM-loss accumulation in cultures of DT40 *Aicda*$^{-/-}$ ΔΨVλ B cells complemented with AID variants-ires-GFP or empty vector (Ctrl). Means (bars) of the median values (dots) obtained from 2 to 5 independent experiments (12–24 cultures/experiment), each normalized to the median value of AID. **d** Class switch recombination activity in *Aicda*$^{-/-}$ mouse primary B cells cultured with LPS and IL-4 and transduced with AID variants-ires-GFP or empty vector (Ctrl). Means (bars) proportion of IgG1+ cells in the GFP+ population at 72 h after transduction for each mouse (dots) from 5 to 7 independent experiments are shown, normalized to AID. In **b–d**, WB of cell extracts probed with anti-AID antibody and loading control are shown at the bottom. **e** Catalytic kinetics of purified recombinant wt AID and R-mutants assayed by the standard alkaline cleavage assay for deamination. (Top) Representative assays used to measure specific activity of wt AID and R-mutants. (Bottom) Mean ± s.d. of four independent experiments were quantified. **f** DNA-binding affinity of wt AID and the R-mutants assayed by EMSA. (Top) Representative EMSA gels are shown. (Bottom) Mean Kd was calculated from four independent experiments for each AID variant. For gel source data, see supplementary Fig. 7

AID, the R-mutants were cytoplasmic in steady state and accumulated in the nucleus after inhibiting nuclear export with leptomycin B (LMB) and/or AID cytoplasmic retention with didemnin B (Did B) (Fig. 2a). Probing for AID by western blot

(WB) in cytoplasmic and nuclear extracts from untreated cells showed that nuclear levels for all AID variants were similar to wt AID, even for R174E that showed reduced cytoplasmic levels compared to wt AID (Fig. 2b). AID can be trapped inside the

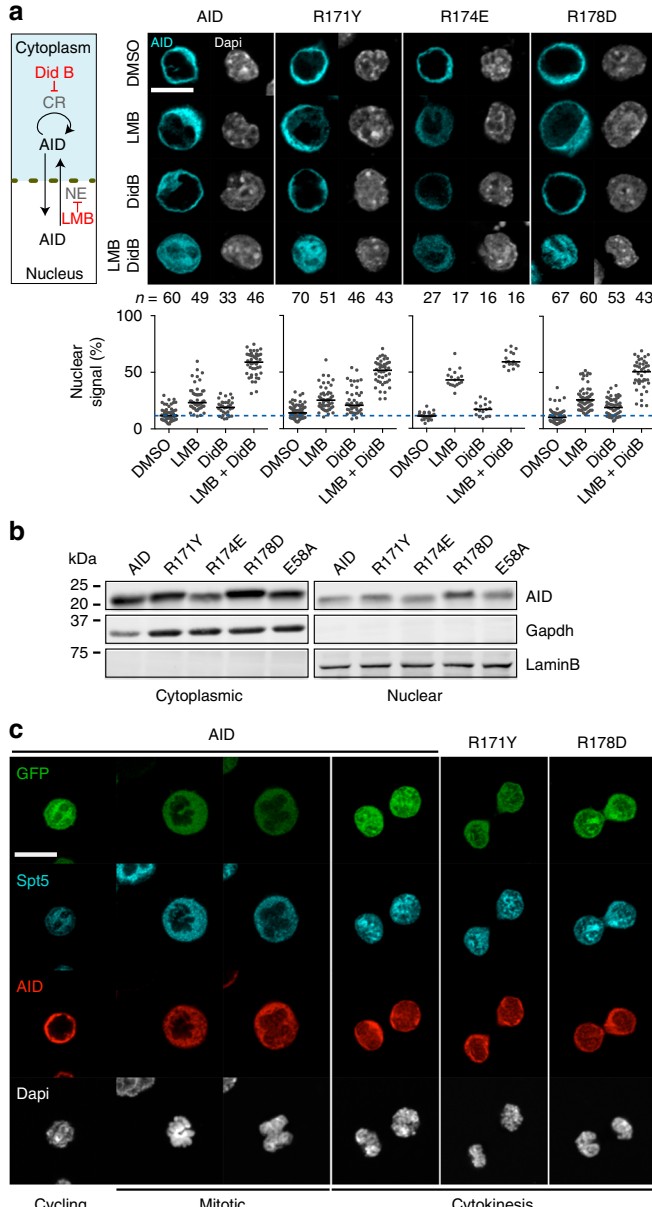

**Fig. 2** Normal nucleocytoplasmic shuttling of AID R-mutants. **a** (Left) Illustration of the mechanisms regulating AID nucleocytoplasmic shuttling. Didemnin B (Did B) inhibits cytoplasmic retention (CR); leptomycin B (LMB) inhibits nuclear export (NE). (Right) Representative confocal microscopy images of AID-deficient CH12 cells transduced with untagged AID variants and analysed by immunofluorescence with anti-AID antibody. Cells were treated for 2 h with DMSO or 10 ng mL$^{-1}$ LMB and/or 100 nM Did B. (Bottom) Proportion of nuclear AID signal for individual cells (dots, *n* indicated above each group). Horizontal bars are mean values. The dotted blue line indicates the mean value of untreated cells expressing wt AID. Compilation of two independent experiments. **b** Representative WBs on cytoplasmic or nuclear lysates from reconstituted AID-deficient CH12 cells probed with anti-AID, -Gapdh and -LaminB antibodies. For gel source data, see supplementary Fig. 7. **c** Representative confocal microscopic images of complemented AID-deficient CH12 cells analysed using anti-AID and anti-Spt5. Cells were determined to be cycling (G1/S/G2), mitotic or in cytokinesis based on DNA condensation and Spt5 access to the DNA. Images are representative of 10–12 observed events per construct from 2 independent experiments. **a**, **c** Magnification 630×. Scale bar, 10 μm

nucleus upon reforming of the nuclear envelope after mitosis[34]. This was maintained by the R-mutants (Fig. 2c and Supplementary Fig. 3d). Thus the mechanisms regulating AID nuclear access are functional in the R-mutants.

As the R-mutants still accessed the nucleus, we asked whether their functional defect was upstream or downstream from deamination. To do so, we measured AID-induced mutations at the Sμ region, the major AID target at the *Igh* locus during CSR[35]. In reconstituted *Aicda*$^{-/-}$ *Ung*$^{-/-}$ B cells, AID R171Y and R178D had substantially lower mutation frequency than wt AID (Fig. 3a), proportionally to the SHM and CSR defects observed (Fig. 1c, d).

We next asked whether this apparent Ig-targeting defect of the R-mutants extended to other loci. As a proxy assay for genome-wide mutagenesis, we measured the decrease in fitness that is associated with AID off-target DNA damage[6,36]. AID expression was sufficient to compromise cell fitness in competitive cultures of CH12 cells, and this effect was enhanced by using a low dose of Did B, which increases the proportion of AID in the nucleus and boosts off-target activity[26] (Fig. 3b). Cells expressing the R-mutants showed no cell fitness defect, just like AID-deficient cells or those expressing catalytically inactive AID E58A (Fig. 3b). To confirm these results, we introduced the R-mutants into the hyperactive AID7.3 variant, which bears three point mutations (outside α6) that increase enzymatic activity three-fold, leading to proportionally higher SHM, CSR, chromosomal translocations and DNA damage in B cells[36,37]. The AID7.3 R-mutants maintained hyperactivity in *E. coli* (Fig. 3c), yet they were still severely deficient for CSR, SHM and cytotoxicity in CH12 and DT40 B cells (Fig. 3d–f). We conclude that the R-mutants enter and accumulate in the nucleus similarly to wt AID but have globally reduced mutagenic activity in B cells.

**The AID RR domain is dispensable for chromatin association.** The nuclear fraction of AID is difficult to visualize in whole cells because of the signal coming from cytoplasmic AID. To test whether the chromatin association of the R-mutants was different from wt AID, we combined a nuclear wash technique[38] with confocal microscopy. We first validated this in situ fractionation method on endogenous AID using CH12 cells, where eliminating the cytoplasmic signal allowed specific detection of AID in the nucleus (Fig. 4a). We then compared the R-mutants to wt AID in transduced AID-deficient CH12 cells (Fig. 4b). Washing away cytoplasm and nucleoplasm, as shown by the loss of the cytoplasmic AID and cell-wide green fluorescent protein (GFP) signals, revealed that AID E58A and the R-mutants showed the same chromatin association as wt AID (Fig. 4b).

To confirm the association of the R-mutants with chromatin, we used a biochemical fractionation protocol that uses an incomplete DNA digestion by micrococcal nuclease (MNase) followed by sequential extractions with increasing salt concentrations[39] (Fig. 4c). As expected, the majority of AID was cytoplasmic, and the lack of cytoplasmic contamination in the isolated nuclei was confirmed by the glyceraldehyde 3-phosphate dehydrogenase (Gapdh) partition (Fig. 4d). Nuclear fractionation showed some Spt5 but no RNAPII in the MNase fraction. All RNAPII and most Spt5 were found in the low and high salt fractions, representing loosely and tightly held transcription complexes, respectively[39] (Fig. 4d). The latter fraction also contained most of the chromatin, judging from nucleosome content (Fig. 4d). Notably, ~60% of the nuclear AID was found in the 600 mM NaCl extract (Fig. 4d, e), indicating tight chromatin association, with little AID in the MNase or 150 mM NaCl fractions. The rest of AID was present in a remaining pellet that was largely devoid of RNAPII, Spt5 or chromatin but contained

Lamin B, thus defining a second pool of nuclear AID that is either part of non-soluble nuclear complexes not directly associated with transcription factors or precipitates during extraction. In either case, the data indicate at least two distinct pools of AID associated with chromatin/nuclear matrix, independently of catalytic activity and biological function. Importantly, the R-mutants had similar distribution profile and proportions as wt AID in all fractions (Fig. 4d, e), showing that they are normally associated with the chromatin in homeostasis. We conclude that the R-mutants dissociate the chromatin interaction from the biological activity of AID, suggesting that these are mechanistically distinct steps.

**AID–chromatin association needs Spt5 but not transcription.** We investigated further the association of AID and the R-mutants to chromatin. Spt5 is important for AID activity in B cells and correlates with AID occupancy genome wide[17], but it is not known whether it is necessary or sufficient for AID–chromatin interaction. Spt5 knockdown in CH12 cells not only reduced CSR

but also overall chromatin association of both endogenous and transduced AID (Supplementary Fig. 4a, b and Fig. 5a). Interestingly, the R-mutants were similarly dependent on Spt5 for chromatin association (Fig. 5a). We directly assessed whether an accumulation of Spt5 at the chromatin was sufficient to induce local AID recruitment by using U2OS cells that contain a genomic Lac operon array (LacO), which is recognized by the Lac repressor (LacR)[40] (Fig. 5b). In this system, a mCherry-LacR-Spt5 fusion recruited AID-GFP, as well as the R-mutants, to the LacO locus (Fig. 5b). As controls, the mCherry-LacR fusion alone did not recruit AID-GFP, and APOBEC1-GFP was not recruited by mCherry-LacR-Spt5.

The association of AID to RNAPII depends on Spt5[17], but whether transcription or RNAPII itself are involved in retaining AID at the chromatin is unknown. To test this, we treated CH12 cells with actinomycin D (Act D), a DNA intercalating compound that disrupts transcription elongation[41]. The eukaryotic RNA polymerases show unequal sensitivity to Act D, RNAPI>RNAPII>RNAPIII[42]. Act D at doses that inhibit RNAPI (0.04 μM) or RNAPI and II (0.4 μM) slightly reduced chromatin-bound RNAPII and caused some AID redistribution to distinct nuclear sites but did not significantly change the amount of AID associated with the chromatin (Fig. 5c). Interestingly, at 4 μM Act D, both RNAPII and AID were depleted from the chromatin (Fig. 5c). A time course at 2 μM Act D showed that RNAPII and AID were concomitantly depleted from the chromatin (Fig. 5d), suggesting that retention of AID at the chromatin might be dependent on RNAPII. The chromatin association of the R-mutants was similarly sensitive to Act D (Fig. 5e).

Surprisingly, AID was partially resistant to chromatin disruption by DNase treatment, which depletes nuclear Spt5 but leaves the nuclear envelope (evidenced by Lamin B staining) intact (Fig. 5f and Supplementary Fig. 4c). In contrast, nuclear AID was largely depleted after RNase treatment, despite chromatin-associated Spt5 being resistant (Fig. 5f).

We conclude that the broad association of AID to chromatin requires RNA and the transcription machinery but not transcriptional activity, with Spt5 accumulation sufficing to promote the recruitment of AID to chromatin, although this

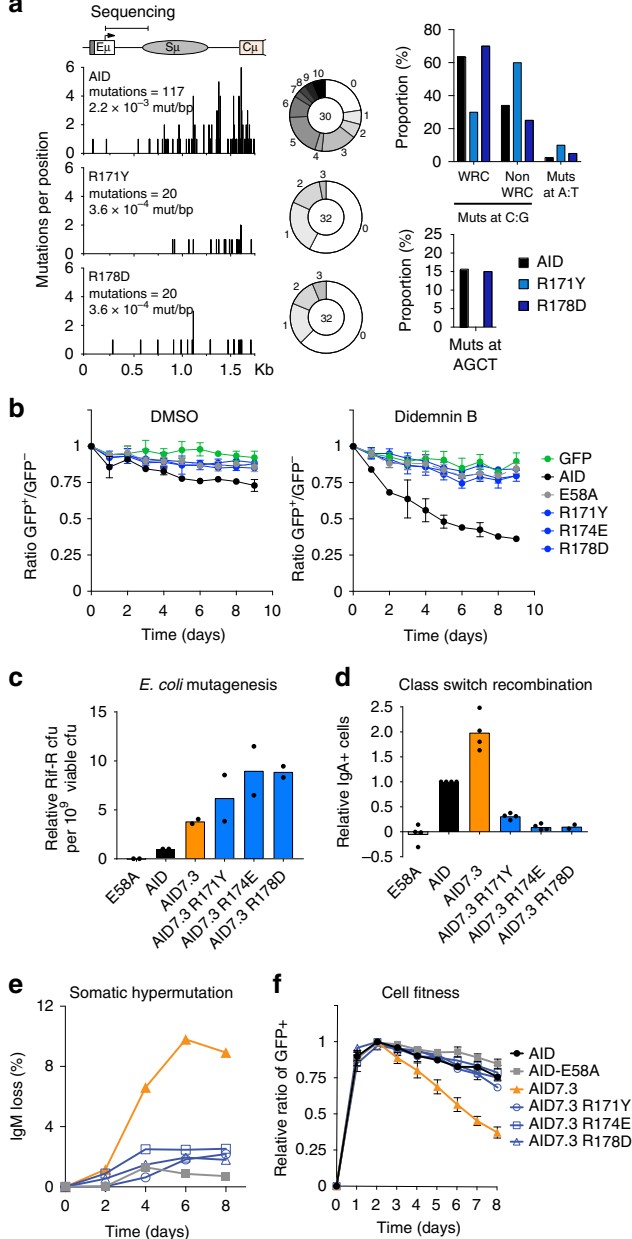

**Fig. 3** The RR domain is necessary for AID *Igh* targeting and for genome-wide DNA damage. **a** (Left) Mutation profiles throughout the Sμ region schematized above, from all analysed sequences. The number and frequency (mutations per base pair) of mutations scored are indicated. (Middle) Pie charts of mutation load per sequence; slices represent proportion of sequences with the indicated number of mutations, with the total number of sequences analysed shown in the centre. (Right) Bar plots of proportion of mutations (Muts) at C:G within WRC (W = A/T, R = A/G) motifs or not or at A:T pairs. **b** Competitive growth of AID-deficient CH12 B cells complemented with AID variants-ires-GFP or empty vector (GFP), co-cultured with untransduced cells (GFP−). Cultures were treated with either DMSO or 1 nM Did B. Mean GFP+/GFP− ratio ± s.d. over time from three independent experiments are shown relative to day 0. **c** Mutagenic activity of AID7.3 variants in *E. coli*, measured by the relative frequency of rifampicin resistance (Rif-R). Means (bars) of median values (dots) from 2 independent experiments (5 cultures/experiment) are shown, normalized to AID. **d** CSR activity of AID or AID7.3 variants-ires-GFP in AID-deficient CH12 B cells, as the proportion of IgA+ cells in GFP+ population (minus background) 72 h post-activation. Means (bars) from 2 to 4 independent experiments (dots), normalized to AID. **e** SHM activity, estimated from IgM-loss over time in DT40 *Aicda*−/− ΔΨVλ B cells complemented with AID variants-ires-GFP, 1 of the 2 independent experiments is shown. **f** Effect of AID or AID7.3 variants-ires-GFP on competitive growth of transduced AID-deficient CH12 B cells. Means GFP+/GFP− ratio ± s.d. over time from three independent experiments, each normalized to maximal value

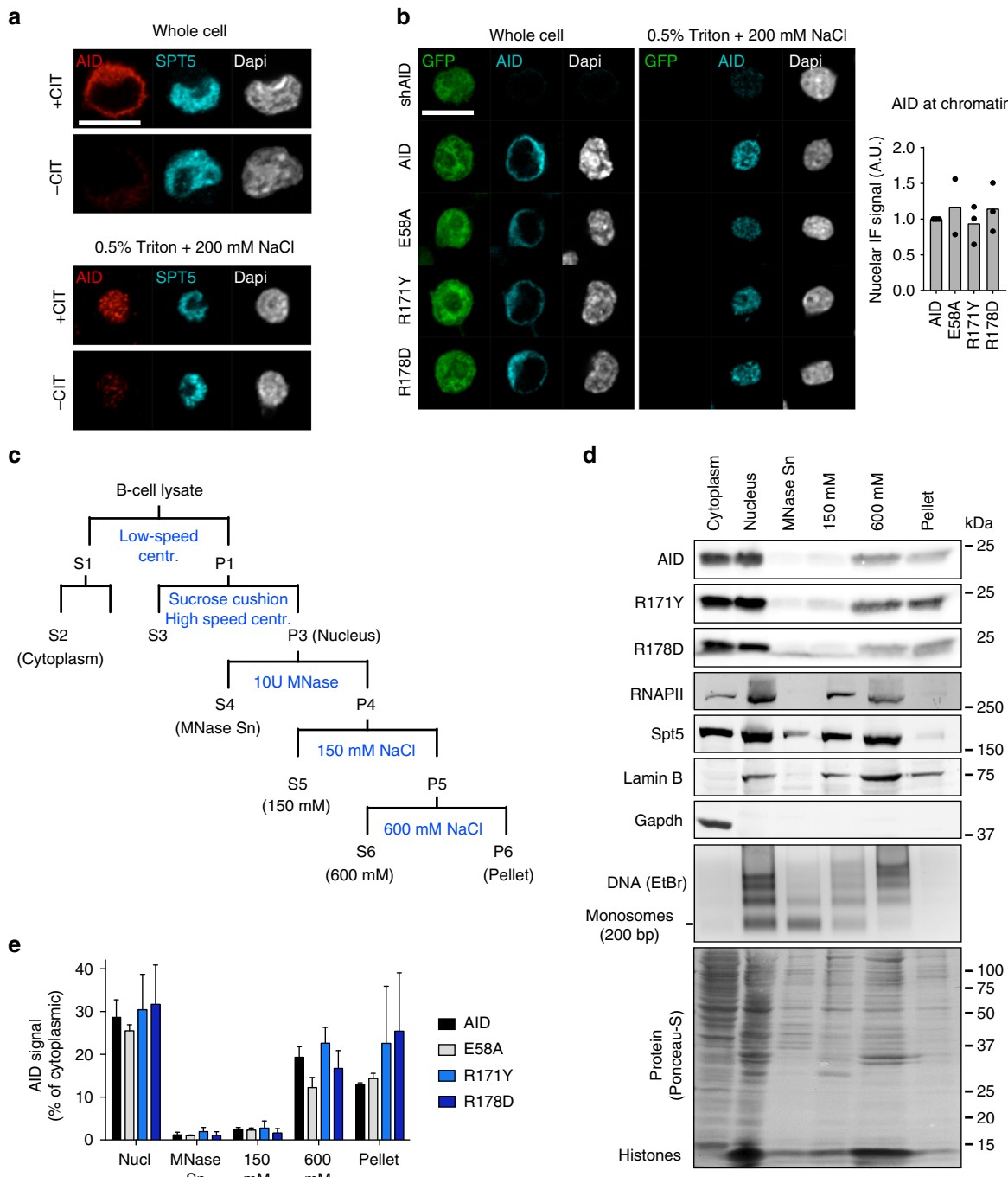

**Fig. 4** The RR domain is dispensable for AID association with tightly held nuclear complexes. **a** Representative confocal microscopic images of CH12 B cells either fixed directly (whole cell) or after nuclear washing (0.5% Triton+200 mM NaCl). Isolated nuclei were analysed by IF to detect endogenous AID and Spt5 and DNA stained by Dapi. Cells were stimulated (+CIT) to induce AID expression or not (−CIT). Representative of three experiments. **b** (Left) Representative confocal microscopic images of GFP, AID (detected by IF) and DNA (Dapi) in reconstituted AID-deficient CH12 B cells, imaged as in **a**. AID and GFP expression were linked via IRES. (Right) Nuclear AID signal was calculated relative to whole-cell intensity for each variant and normalized to the wt AID value. Plotted are means (bars) from 3 independent experiments (dots) with 20–77 cells per condition per experiment. **a**, **b** Magnification 630×. Scale bar, 10 μm. Laser power and/or gain were increased for imaging after nuclear wash (see Methods). **c** Scheme for biochemical fractionation of B cells. Fractions analysed are indicated in brackets. **d** Representative WBs on the indicated fractions from reconstituted AID-deficient CH12 B cells. Antibodies against AID, RNAPII, Spt5, Lamin B and Gapdh were used as indicated. Representative agarose gel with purified DNA stained using ethidium bromide and Ponceau-S staining of total protein are also shown. For gel source data, see supplementary Fig. 7. **e** Quantification of AID signal in each lane from **d**, normalized to the respective cytoplasmic AID. Means + s.d. from three independent experiments

effect could be indirect and likely requires additional factors, such as RNA. The fact that chromatin-associated AID exists in at least two distinct fractions, only one of which contains Spt5 and RNAPII (Fig. 4d), implies that AID is dynamically associated with

these fractions. Constant cycling through Spt5 would explain why Spt5 knockdown evicts AID from the chromatin. The RR domain is dispensable for this dynamic interaction, yet necessary for function, suggesting that it mediates productive targeting of AID.

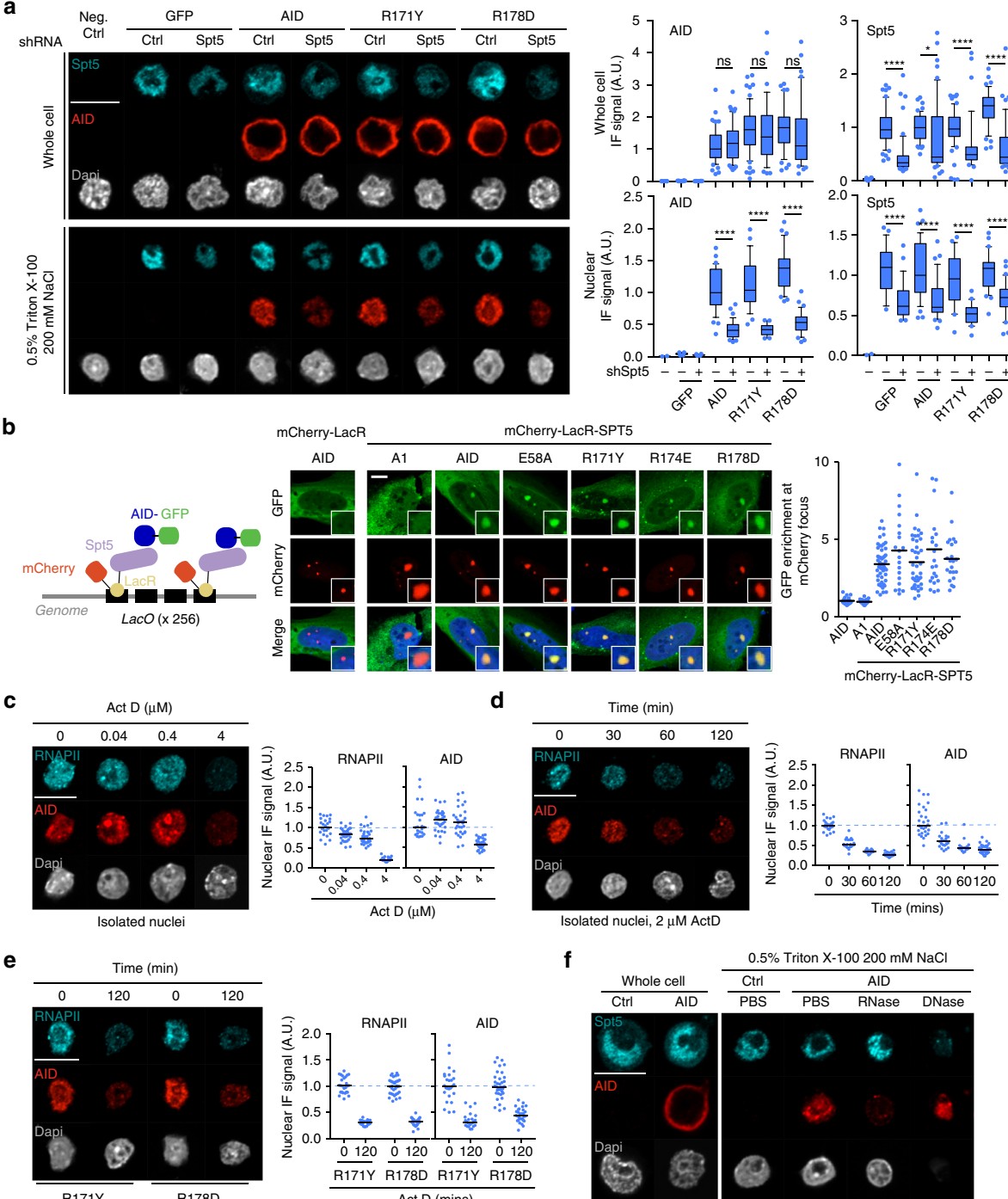

**Fig. 5** Requirements for chromatin association of AID. **a**, **c**–**f** Representative confocal microscopic images of GFP, immunofluorescence (for AID and Spt5 or RNAPII) and DNA staining (Dapi) in reconstituted AID-deficient CH12 B cells. Nuclear wash was carried out as in Fig. 4. Magnification 630×. Scale bar, 10 μm. **a** (Left) CH12 cells were transduced with either a shRNA control or against Spt5 as indicated. (Right) Box plots (25–75 percentiles with median and 10–90 percentiles whiskers of AID and Spt5 signal from individual whole cells or isolated nuclei, normalized to the median of wt AID. One of the two independent experiments is plotted (21–66 cells per condition). Differences were evaluated by unpaired, two tailed *t*-test (ns, not significant, *P < 0.05, ****P < 0.0001). **b** (Left) Schematic of the *lacO*/LacR system. (Middle) Representative confocal microscopic images of GFP or mCherry and their overlay, including DNA (Dapi). Small boxes highlight typical mCherry foci. (Right) Quantification of GFP signal at each mCherry focus, normalized to non-focus nuclear GFP signal (dots). The mean for each construct (bars) is shown. Compilation of two independent experiments. Magnification 400×. Scale bar, 10 μm. **c** As in **a**, but cells were treated for 60 min with DMSO (0) or the indicated dose of Actinomycin D (Act D) prior to plating and washing. **d** As in **a**, but cells were treated with DMSO (*t* = 0) or for various times with 2 μM Act D prior to plating and washing. **e** As in **a** for cells expressing AID R-mutants treated with either DMSO or 2 μM Act D for 120 min. **c**–**e** Average AID and RNAPII signal for individual nuclei (dots), with bars indicating population median, normalized to the median of untreated cells (dotted line), with 17–32 cells per condition from 1 experiment. **f** As in **a** but nuclei were incubated at 37 °C with PBS control, RNAse or DNase during nuclear wash. Images are representative of 42–78 cells per condition and 2 independent experiments. IF immunofluorescence, A.U. arbitrary units

**Modular nature of the RR domain**. To obtain mechanistic insight into the defect of the AID R-mutants, we asked whether a wt AID RR domain could rescue their function by fusing either wt AID or the R-mutants to the catalytically inactive AID E58A mutant. The control AID-AID E58A fusion protein had reduced mutagenic activity in *E. coli*, compared to the AID monomer, but still produced substantial CSR (Fig. 6a). The analogous R-mutant

fusions had the same activity as AID-AID E58A not only in *E. coli* but also for CSR (Fig. 6a). In this experiment, catalytic activity is derived from the R-mutant AID, while AID E58A provides the RR domain function. This was confirmed by further mutating the RR domain of AID-E58A in these fusions, which did not affect activity in *E. coli* but eliminated CSR activity (Fig. 6a). This demonstrates that the two functions are modular and suggests

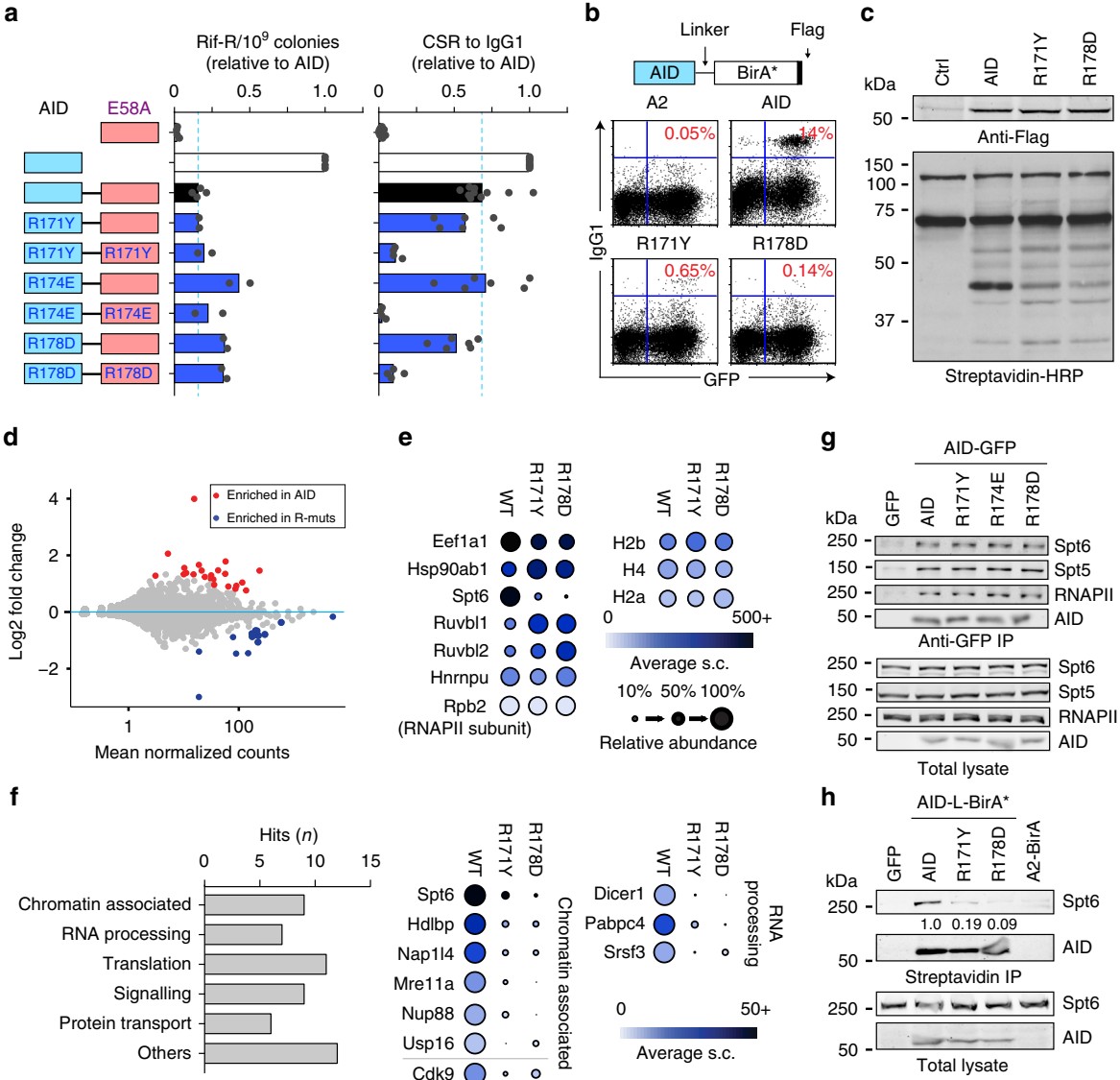

**Fig. 6** Protein–protein interactions of AID and R-mutants. **a** (Left) schemes of AID, AID E58A and fusions thereof indicating additional mutations. (Middle) Relative ability to produce rifampicin resistant (Rif-R) *E. coli* colonies is shown by means (bars) of median values (dots) from 2 to 4 independent experiments (5 clones/experiment), normalized to the median of AID from each experiment. (Right) Relative CSR capacity of the same proteins expressed in *Aicda*$^{-/-}$ mouse B cells. Means (bars) from 4 to 10 independent mice (dots) from 2 to 5 independent experiments are shown, normalized to AID. **b** Schematic of the AID fusions to BirA* and representative flow cytometric plots for CSR, the proportion of IgG1+ cells in the GFP+ population, in reconstituted *Aicda*$^{-/-}$ mouse B cells. APOBEC2 (A2) was used as control. **c** Representative WB of the AID-BirA* fusions (anti-flag) and biotinylated targets (streptavidin) from reconstituted *Aicda*$^{-/-}$ mouse B cells after incubation with biotin. Untransduced cells were used as control (Ctrl). **d** MA plot showing the fold change in spectral counts (s.c.) as a function of average s.c. of all hits for wt AID versus R-mutants. Red and blue dots represent significant differences identified according to negative binomial distribution by the DESeq2 software. **e** (Left) Circle plots of BioID signal for selected known AID interacting partners. Circle size indicates relative abundance normalized to the variant with the most s.c. Circle colour represent actual s.c. Scales are shown. **f** (Left) Broad functional categories of BioID associations found reduced in the R-mutants compared to wt AID. (Right) Circle plots for hits in the indicated categories. **g** Anti-GFP pull down from whole-cell lysates of AID-deficient CH12 B cells reconstituted with AID variant-GFP fusions. Spt6, Spt5, RNAPII and AID were detected by WB from immunoprecipitates or lysates. Representative of two independent experiments. **h** Streptavidin pull down of biotinylated proteins from whole-cell lysates of AID-deficient CH12 B cells reconstituted with AID variant-BirA* fusions after 24 h pulse of biotin. Spt6 and AID were detected by WB from either pull downs or lysates. Ratios of Spt6 to AID signal normalized to AID-BirA* are indicated. Representative of two independent experiments. For gel source data, see supplementary Fig. 7

that the R-mutants are still intrinsically capable of deaminating the *Ig* locus.

**AID R-mutants lose specific interactions in vivo**. Our results suggest that the RR domain may mediate an interaction necessary for targeting AID activity. To identify proteins that associate with AID in a manner that depends on the intact RR domain, we compared the interactome of AID and two R-mutants in live B cells. We used BioID, a proximity-based biotin labelling technique in which a bait is fused to the promiscuous BirA* biotin ligase that can label the protein environment of the bait in an ~10 nm radius[43]. We generated an AID-BirA* fusion, which was active for CSR in transduced *Aicda*[−/−] B cells, and its R171Y or R178D derivatives (Fig. 6b). Addition of biotin to the cultures led to biotinylation of proteins in cells expressing BirA* fusions (Fig. 6c), proportionally to AID-BirA* expression levels (Supplementary Fig. 5a, b). The vast majority of AID interactions were conserved in the R-mutants, including many validated AID interactors (Fig. 6d, e and Supplementary Fig. 5f), further confirming the structural integrity of the mutants. Using four different statistical methods, we identified 54 proteins that showed reduced interaction with both R-mutants compared to AID, 29 of these were identified by at least two methods (Fig. 6d, Supplementary Fig. 5c–e, and Supplementary Table 1, see Methods). Functional annotation of the BioID hits showed nine chromatin-associated factors, including the histone chaperones Spt6 (Supt6h) and Nap1l4 as well as the P-TEFb component Cdk9, which are functionally linked to transcription elongation[44] (Fig. 6f). We also found seven factors linked to co-transcriptional RNA processing: splicing and mRNA transport factors, as well as Dicer which can bind to dsRNA resulting from convergent transcription induced by R-loops[45]. Of these, only Spt6 has been previously shown to interact with AID by co-immunoprecipitation (co-IP). Interestingly, standard pull down from cell extracts showed that GFP-tagged R-mutants still co-IP Spt6 (Fig. 6g). However, streptavidin pull down of proteins biotinylated in live CH12 B cells expressing AID- or R-mutants-BirA* confirmed the lack of interaction with the R-mutants in live cells (Fig. 6h). These data together strongly suggest that the BioID result reflects the loss of the functional interaction between the R-mutants and Spt6, rather than their physical inability to form a complex with Spt6. We conclude that the defect in the R-mutants lies after chromatin association but prior to the transcriptional step in which Spt6 is recruited.

**AID R-mutants fail to occupy the target genes body**. The R-mutants were able to interact with RNAPII (Fig. 6e, g), yet did not mutate B cells. As Spt5 is recruited to promoter-proximal paused RNAPII, and Spt6 only to active transcription[46], we hypothesized that the R-mutants might fail to progress from paused to elongating RNAPII. We therefore compared the occupancy of wt AID and the R-mutants near the transcription start site (TSS) versus the gene body at the physiological and one prominent off-targets of AID. We used an antibody against the E5 domain of AID to perform ChIP from *Aicda*[−/−] mouse B cells complemented with either wt AID or the R-mutants and stimulated for CSR to IgG1. We first analysed the Sμ-region, the major AID target at the *Igh* locus during CSR[35], comparing AID occupancy at the region around the TSS where paused RNAPII is expected[44], to the downstream Sμ region (Fig. 7a). ChIP showed that wt AID was present in all amplicons but was highest at the Sμ region (Fig. 7a), where elongating RNAPII was previously shown to stall[47]. This profile was not an artefact of AID overexpression, as endogenous AID had a similar distribution (Fig. 7b). Spt6 was also present in all amplicons (Fig. 7b), as expected for a highly transcribed

gene[46]. In contrast, the R-mutants were present at the TSS but depleted from the Sμ gene body (Fig. 7a), and we obtained the same result at the Sγ1 region (Fig. 7b). Moreover, we were able to detect low but reproducible ChIP signals for AID at the *IL4Ra* locus, a known AID off-target[48], where the R-mutants were specifically depleted within the gene body but showed normal occupancy at the promotor, compared to wt AID (Fig. 7d). As a control, no AID or R-mutants were detected at either the promoter or the gene body of *Gapdh* (Fig. 7e), which is not occupied by endogenous AID[48]. To test whether the same defect could underlie the inability of the R-mutants to do SHM, we analysed AID occupancy at the IgV region in complemented DT40 *Aicda*[−/−] ΔΨVλ B cells. Again, the R-mutants were equally recruited to the IgV promoter but substantially depleted from the gene body, compared to wt AID (Fig. 7f). As an additional control in all ChIPs, we used AID E58A, which showed similar occupancy as wt AID at all at the regions tested, indicating that the simple lack of deamination ability could not explain the observed occupancy profile. All AID variants showed similar expression levels in B cells (Fig. 7g).

We conclude that the R-mutants can occupy the promoter-proximal regions of all physiological and at least one off- AID targets, yet fail to occupy the corresponding gene bodies. Thus, the lack of association between the R-mutants and Spt6 most likely reflects the uncoupling of the AID R-mutants from elongating and/or stalled RNAPII, which suggests the existence of a licensing step for productive targeting of AID.

**Constitutively nuclear AID bypasses licensing**. The RR domain was dispensable for mutagenesis in *E. coli*, which also depends on transcription[49], thus suggesting that the licensing step can be bypassed. We therefore asked whether a constitutively nuclear AID might rescue the defect of the R-mutants by mimicking the *E. coli* situation, where proximity to the genome is not regulated. We introduced the R-mutants into AIDΔE5, a nuclear AID variant[24,36] that results in >3-fold higher steady-state levels of nuclear enzyme in CH12 cells (Fig. 8a). The AIDΔE5 R-mutants were as active as AIDΔE5 in *E. coli* (Fig. 8b). In contrast to their effect on full-length AID, mutations R174E and R178D did not prevent the detrimental effect on cell fitness or ability to generate γH2AX foci of AIDΔE5 in B cells (Fig. 8c, d). Moreover, when expressed in *Aicda*[−/−] DT40 B cells, AIDΔE5 R174E and AIDΔE5 R178D produced similar SHM levels than AIDΔE5 at the IgV, as judged by the IgM-loss assay as well as by mutation frequency and profile (Fig. 8e, f and Supplementary Fig. 6). While AIDΔE5 is deficient for CSR, it can target and deaminate the Sμ[36]. AIDΔE5 R174E deaminated the Sμ with the same frequency and profile as AIDΔE5 in complemented *Aicda*[−/−] *Ung*[−/−] primary B cells, with R171Y and R178D producing less but still substantial mutations (Fig. 8g). Interestingly, AIDΔE5 R171Y consistently produced less SHM, DNA damage and Sμ mutations than AIDΔE5 and the other two mutants, despite equal expression (Fig. 8h), suggesting an additional role of this residue for mutagenesis in B cells. Nonetheless, the results with the other two mutants indicate that at least R174 and R178 are dispensable for deaminating the physiological AID targets when the proposed licensing step is bypassed, which can be achieved when nuclear AID levels are not limiting. We conclude that any possible deamination defect in the specific R-mutants used here are insufficient to explain their biological defects and that limiting AID nuclear levels helps enforcing licensing.

**Discussion**

The genes that are mutated by AID in B cells share a transcriptional landscape associated with super-enhancers and convergent

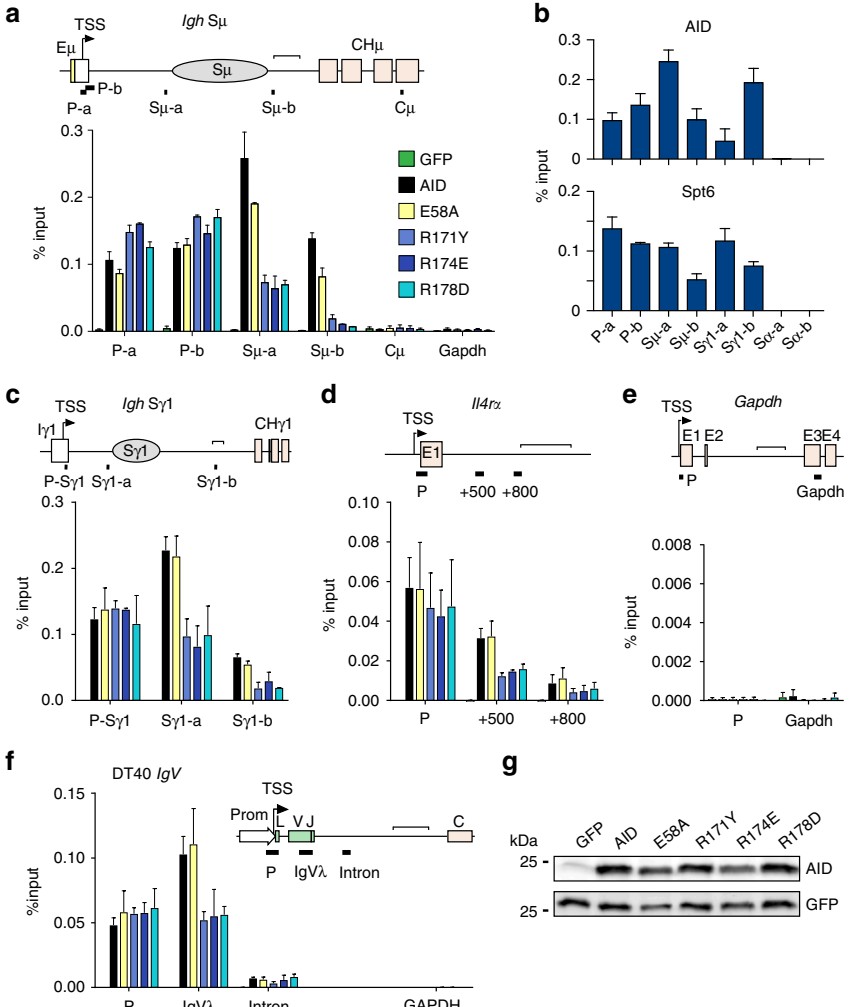

**Fig. 7** AID R-mutants are uncoupled from transcription elongation. **a**, **c–e** ChIP qPCR results using anti-AID antibody on extracts from reconstituted *Aicda* $^{-/-}$ mouse B cells activated with LPS and IL-4. Means + s.e.m. from two independent experiments are plotted. A scheme of each genomic region analysed is shown at the top of each panel with the qPCR amplicons indicated. **b** ChIP qPCR for endogenous AID and Spt6 from wt mouse B cells treated and analysed as in **a**, including amplicons in the non-transcribed Sα region. Means + s.d. from three independent experiments are plotted. **f** (Bottom) ChIP qPCR for AID from reconstituted DT40 *Aicda* $^{-/-}$ ΔΨVλ B cells. Means + s.d. from three independent experiments are plotted. An additional amplicon within *Gapdh* was used as a negative control. **g** WB of cell extract from reconstituted *Aicda* $^{-/-}$ mouse B cells, probed with antibodies recognizing GFP and AID. For gel source data, see supplementary Fig. 7

transcription[13–15]. This demonstrates that AID targeting is based on the local chromatin architecture, rather than the fixed features of the target genes. The mechanism underlying the productive targeting of AID remains unclear, as it cannot be explained by differential recruitment of AID given that AID occupancy is insufficient for mutation[4,19,20,48]. Here we uncover an obligatory licensing step for AID that provides insight into this fundamental question.

We show that specific substitutions of either Arg residues 174 or 178, and to a lesser extent Arg 171, dissociate the ability of AID to associate with chromatin and occupy promoter-proximal regions from its ability to induce mutations in the downstream genes in B cells. It was recently proposed that these Arg in AID α6 form part of an assistant patch required for deaminating structured DNA, especially G4 DNA that forms at the S-regions[50]. Several lines of evidence indicate that the functional defect of our R-mutants cannot be explained by a potential defect in deaminating structured DNA: (1) None of the R-mutants affect *E. coli* genome mutagenesis or its ability to bind to and deaminate a bubble substrate in vitro (Fig. 1b, e, f); (2) Catalytically inactive

AID E58A can occupy the target regions; (3) When licensing is bypassed using the AIDΔE5 variant, R174E has no effect on, and R178D and R171Y only reduce by 2–3-fold, the ability to mutate the Sμ (Fig. 8g), a much less dramatic effect than these mutations have on CSR activity. In fact, AID R174E, which has virtually no CSR activity (Fig. 1d and ref. [50]), retains at least 50% enzymatic activity on G4 DNA[50]; (4) the IgV does not form G4 DNA[51] and AIDΔE5 R174E and AIDΔE5 R178D show normal SHM (Fig. 8f); (5) These nuclear AID mutants also produce normal levels of off-target DNA damage in B cells (Fig. 8d). On the other hand, AIDΔE5 R171Y is less efficient in deaminating the Sμ and even more impaired for SHM and DNA damage, indicating an additional defect compared to R174E and R178D. This defect could be biochemical but remains to be studied. Thus our results do not dispute the involvement of the α6 Arg residues in assisting deamination of G4 DNA[50] but provide good evidence that they have an additional function, which we propose is AID licensing.

AID and the R-mutants broadly associate with chromatin. We show that Spt5 accumulation can promote AID recruitment to chromatin, fitting the observation that AID accumulates at

genomic Spt5 peaks[17,48], although this could be indirect and other factors might share this capacity. Several factors seem to contribute to the association of AID with chromatin. The physical presence of RNAPII may be required, but transcription itself is dispensable. Our data indicate that Spt5 is required to maintain chromatin-wide association of AID, as Spt5 depletion evicts AID from chromatin. On the other hand, RNase treatment also evicts AID from the chromatin but does not evict Spt5. Moreover, a large proportion of chromatin-associated AID is in a fraction devoid of RNAPII or Spt5, as shown by salt fractionation. One

possible explanation for these observations is that the broad distribution of AID at the chromatin reflects dynamic associations or shuttling between Spt5 complexes and at least another distinct RNA-dependent complex. We propose that dynamic association with chromatin and accumulation at Spt5-rich regions allow AID to sample multiple loci. Similar diffusion and accumulation behaviour has been described for transcription factors searching for their target DNA sequences[52]. Unlike transcription factors, AID does not recognize a specific DNA sequence[2], and the licensing step we report would provide the extra level of

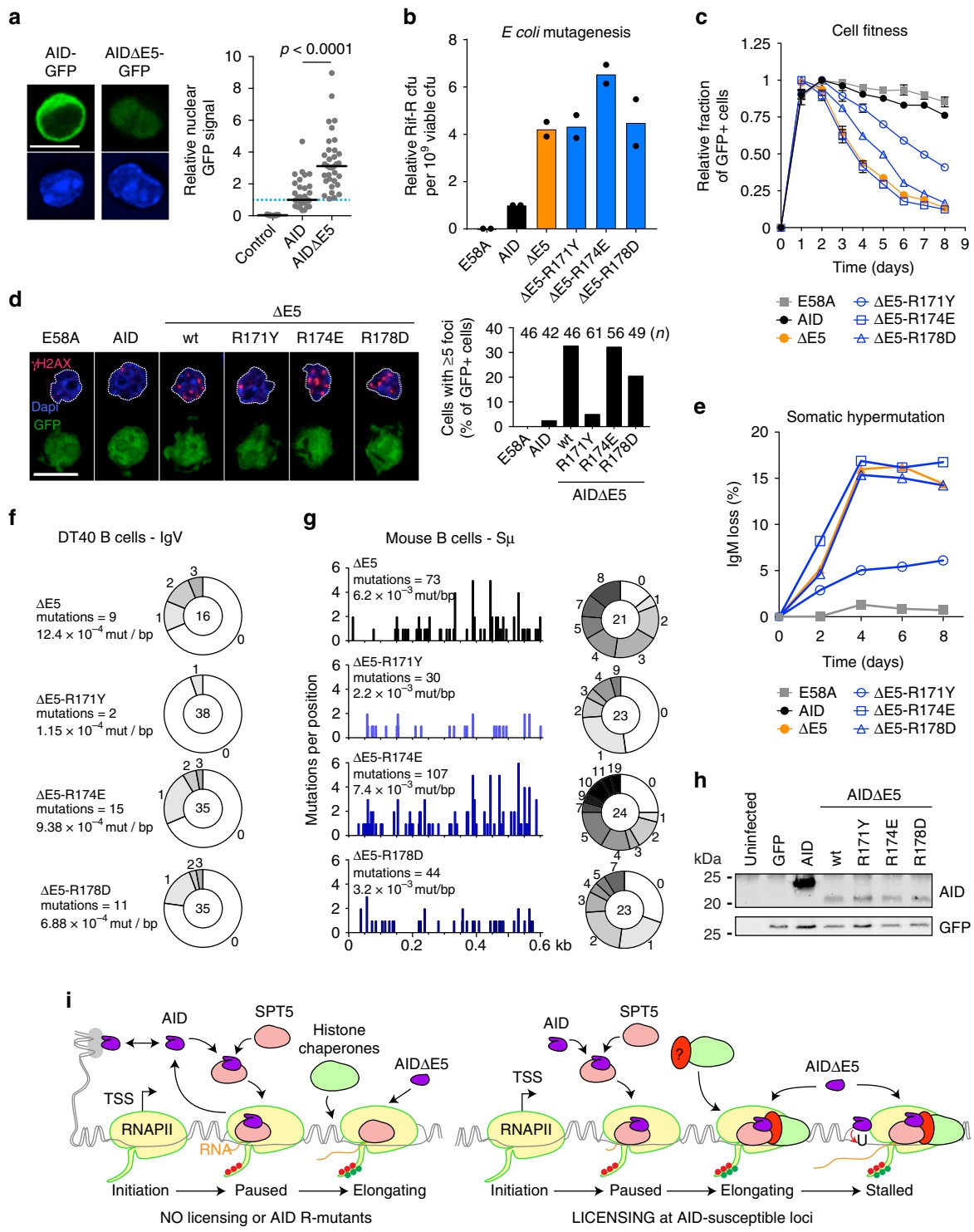

regulation to recognize deamination targets after associating with Spt5-rich promoters.

The current model for AID targeting posits that AID mutates when RNAPII stalls, but this does not explain how AID associates with stalled polymerase, beyond invoking a role for Spt5[53]. Like wt AID, the R-mutants associate with Spt5 and are recruited to the promoter-proximal region of various target sites (Fig. 7) but fail to act, thereby uncovering an additional level of regulation. Our results are consistent with AID being recruited to the TSS and progressing with transcription elongation until the polymerase becomes stalled, rather than being directly recruited to stalled polymerases. Abnormally high nuclear AID levels can bypass this licensing step, possibly by direct recruitment to stalled polymerases, resulting in a highly mutagenic enzyme. This would provide a rationale for the abundance of mechanisms that restrict AID nuclear access[1], which would serve to enforce licensing.

In conclusion, our data are consistent with a model (Fig. 8i) in which AID samples the genome by dynamic and broad chromatin association. The low concentration of AID in the nucleus would favour its accumulation at promoter-proximally paused RNAPII, probably via Spt5[17,48], but likely involving additional factors. Only those loci within a permissive transcriptional landscape would contain the factors that mediate the coupling of AID to transcription elongation, licensing AID to deaminate the downstream region, as shown for the IgV and Sμ regions. The R-mutants fail to interact with Spt6 in live cells but have not lost biochemical interaction ability with this factor, suggesting that licensing takes place before their encounter during transcription, rather than Spt6 being directly involved. Elucidating the molecular mechanism of licensing will require considerable work but it is likely that it takes place within super-enhancers[13–15]. Our result at Il4ra suggests that licensing permits mutation of AID off-targets (which will need to be confirmed genome wide) but is probably more active at the Ig loci and absent from those loci that are not mutated by AID. We speculate that the nuclear deaminase APOBEC3B and other AID paralogues with antiviral activity[54] have lost the RR domain and thereby licensing to prevent mutating the self-genome during transcription.

## Methods

**Animals.** Aicda[−/−] mice[3] (a gift from Dr. T. Honjo, Kyoto University, Japan) and Aicda[−/−] Ung[−/−] mice (Ung[−/−] mice[55] were a gift of Dr H. Krokan, Norwegian University of Science and Technology, Norway) in C57BL/6J background were bred at the specific pathogen-free facility of the Institut de Recherches Cliniques de Montreal. Male and female mice of 6 weeks to 8 months old were used as a source of B cells. Animals were euthanized in $CO_2$ chamber. All animal work was approved by the IRCM animal protection committee (AUP 2015-10) in accordance to the guidelines of the Canadian Council for Animal Care.

**DNA constructs.** Retroviral vector pMXs human AID-, AIDΔE5- and AID7.3-ires-GFP, as well as mouse AID-GFP, have been described[24,26,36]. Human AID fusions were assembled as BamHI-AID-EcoRI-Linker-HindIII-AID-E58A-XhoI cassettes into pTrcHisA (ThermoFisher) or pMX-ires-GFP (Cell Biolabs). The linker encoded a flexible (SGGGG)x3 peptide. Human SPT5 was PCR amplified and cloned into a gateway-compatible mCherry-LacRep destination vector[56] (a gift from Dr. D. Durocher, University of Toronto, Canada). Mouse AID and Linker-BirA* were PCR amplified using gateway-compatible primers and cloned into appropriate donor vectors to generate AID-BirA* fusions into a homemade gateway-compatible pMX-ires-GFP bearing a gateway cassette cloned BamHI-EcoRI by using Multisite gateway technology (Invitrogen). AID variants were generated by PCR amplification with ad hoc oligonucleotides or by quick-change site-directed mutagenesis using KOD1 DNA polymerase (Toyobo Inc.). For oligonucleotide sequences, see Supplementary Table 2.

**Cell culture and transduction.** CH12 cells[3] (A kind gift from Dr T Honjo, Kyoto University, Japan) and primary B lymphocytes were cultured in RPMI 1640 media (Wisent) at 37 °C with 5% (vol vol$^{-1}$) CO2. U2OS lacO cells[40] (Obtained from Dr A Orthwein, Lady Davis Institute, Montreal) were cultured in McCoy's 5a media (Wisent). Media were supplemented with 10% foetal bovine serum (Wisent), 1% penicillin/streptomycin (Wisent) and 0.1 mM 2-mercaptoethanol (Bioshop). DT40 Aicda[−/−] ΔΨVλ cells[57] (A kind gift from Dr H Arakawa, IFOM, Italy) were cultured in RPMI 1640 supplemented as above plus 1% chicken serum (Wisent). CH12 cells stably expressing short hairpin RNA against AID have been described[58]. All cell lines were regularly tested for mycoplasma, and cell line identity was inferred from relevant functional assays. Naive splenic B cells from Aicda[−/−] or Aicda[−/−] Ung[−/−] mice were isolated from total splenocytes depleted with anti-CD43 microbeads (Miltenyi Biotec, Cat.#130-049-801) in an autoMACS cell separator (Miltenyi Biotec). For retroviral complementation of DT40 or CH12 cells, VSV-G, MLV gag-pol and pMXs vectors (1:1:4 ratio, 2.5 μg DNA total) were transfected into HEK293 cells using Trans-IT LT-1 (Mirus Bio, Cat.# MIR 2305). HEK293 were chosen because of their high transfection efficiency and virions' production confirmed in test infection assays. Retrovirus for primary cell transduction was produced using Plat-E ecotropic packaging cells (A kind gift of Dr. T Kitamura, University of Tokyo, Japan)[59] transfected with pMXs vectors. For infections, 1 mL of HEK293 or 1.5 mL Plat-E supernatant at 48 h post-transfection was used to infect $10^6$ B cells in 24-well plates, in the presence of 8 μg mL$^{-1}$ polybrene (Sigma, Hexadimethrine bromide Cat.# H9268), by spinning at $600 \times g$ for 90 min at 32 °C. Medium was replaced 4 h later.

**Reagents and antibodies.** Stock aliquots: 50 μg mL$^{-1}$ LMB in ethanol (LC Laboratories), 20 mM Didemnin B in dimethyl sulphoxide (NSC 325319; provided by the Natural Products Branch, National Cancer Institute, Bethesda, MD) and 2 mM Actinomycin D in DMSO (Santa Cruz Biotechnology). Drugs were kept at −20 °C in the dark and diluted fresh before each experiment. Antibodies used for immunofluorescence (IF): for human AID; rat monoclonal antibody (mAb) anti-AID (1:500, EK2 5G9, Cell Signaling), for mouse AID; rat mAb anti-AID (1:250, mAID-2, eBioscience), rabbit mAb anti-SPT5 (1:500, EPR5145(2), Abcam), rabbit anti-RNAPII (1:100, H-224, SC Biotechnology), goat anti-LaminB (1:500, M-20, SC Biotechnology), Rabbit anti-γH2AX (1:800, #2577, Cell Signaling). With the exception of anti-rat Dylight 550 (1:500, SA5-10027, ThermoFisher Scientific), all

**Fig. 8** Nuclear exclusion is necessary to enforce AID licensing. **a** (Left) Representative confocal microscopic images of AID-GFP and AIDΔE5-GFP fusions in CH12 B cells. (Right) Nuclear GFP signal was measured as the overall GFP signal overlapping with a nuclear mask, generated using Dapi signal. Cells expressing each construct were fixed and imaged with identical settings in parallel. Individual cell values (dots) are plotted relative to the median of AID-GFP, from one experiment. Control shows untransfected cells signals. Differences were evaluated by unpaired, two tailed t-test. **b** Relative mutagenic activity of AIDΔE5 variants in E. coli, measured as the frequency of rifampicin-resistant (Rif-R) colony-forming units (cfu). Means (bars) of medians (dots) from 2 independent experiments (5 cultures/experiment), normalized to AID. **c** Effect of AID or AIDΔE5 variants-ires-GFP on the competitive growth of transduced AID-deficient CH12 B cells. Means GFP$^+$/GFP$^-$ ratio ± s.e.m. over time from two independent experiments, each normalized to maximal value. **d** (Left) Representative confocal microscopic images of GFP and γH2AX immunofluorescence with DNA staining (Dapi) in Aicda[−/−] mouse B cells activated with LPS and IL-4 complemented with AID variants-ires-GFP vectors. (Right) The proportion of cells with ≥5 γH2AX foci per nucleus is plotted from 1 experiment, with n cells counted. **a, d** Magnification 630×. Scale bar, 10 μm. **e** SHM capacity of AIDΔE5 variants-ires-GFP measured by IgM-loss over time in complemented DT40 Aicda[−/−] ΔΨVλ B cells. One of the two independent experiments is shown. **f** Number and frequency (mutations per base pair) of mutations scored at the IgV of DT40 Aicda[−/−] ΔΨVλ B cells expressing AIDΔE5 variants obtained from GFP+ cells sorted at day 3 post-transduction (see **e**) from one experiment. Mutation load pie charts, with slices representing proportion of sequences with the indicated number of mutations and total sequences analysed indicated in the centre. **g** (Left) Mutation profiles and (right) mutation load, as in **f** scored at the Sμ region of Aicda[−/−]Ung[−/−] mouse B cells transduced with AIDΔE5 variants. **h** WB of cell extracts from reconstituted mouse B cells probed with antibodies recognizing GFP and the N-terminus of AID. For gel source data, see supplementary Fig. 7. **i** Model for AID targeting (see Discussion)

other IF secondary Abs were AlexaFluor conjugates (1:500, invitrogen): anti-rat-680, anti-rabbit-546, anti-rabbit-680, and anti-goat-680. Antibodies use for WB: rat mAb anti-AID (for human AID; 1:1000, EK2 5G9), a 1:1 mixture of mouse mAb anti-AID 52–1 and 39–1 (1:5000) specific against unmapped epitopes within human AID N-terminal half (a gift from Dr. M. Neuberger, Cambridge, UK; validated in ref. [36]), rat mAb anti-AID (for mouse AID; 1:500, mAID-2 eBioscience), rabbit anti-Actin (1:3,000, A2066, Sigma), rabbit anti-GFP (1:2000, 11122, Invitrogen), rabbit anti-GFP-HRP (horseradish peroidase) (1:5000, 130-091-833, Miltenyi Biotec), mouse mAb anti-GAPDH (1:3000, H-12, SC Biotechnology), goat anti-LaminB (1:2000, M-20), rabbit anti-SPT5 (1:500, H-300, SC Biotechnology), rabbit anti-RNAPII (1:500, H-224), rabbit anti-SPT6 (1:2000, A300-801A, Bethyl Laboratories), Streptavidin-HRP (1:10,000, N100, Thermo Scientific). Secondary antibodies were anti-mouse-, anti-rat-, anti-goat- or anti-rabbit-AlexaFluor680 (1:10,000; Invitrogen), read using an Odyssey CLx apparatus (Li-COR), or the ChemiDoc XRS+ Imaging system (Biorad) for HRP conjugates developed by chemiluminescence (34080, Thermo Scientific).

**Monitoring SHM and CSR.** SHM was measured by fluctuation analysis of IgM-loss in DT40 $Aicda^{-/-}$ $\Delta\Psi V\lambda$ cells[57] complemented by retroviral transduction with AID or mutants thereof, as described[36]. Briefly, cells were transduced with pMX AID variant-ires-GFP vectors, 200 GFP+ cells were sorted and expanded in 24-well plates for 3 weeks before measuring the proportion of IgM-loss by flow cytometry using anti-IgM conjugated with R-phycoerythrin (clone M-1, SouthernBiotech). Gating strategy is shown in Supplementary Fig. 8a. For AID7.3 and AIDΔE5, IgM-loss was measured in bulk every 2 days for 8 days to determine SHM kinetics. CSR to IgG1 in complemented mouse $Aicda^{-/-}$ splenic B cells was induced by adding 5 µg mL$^{-1}$ lipopolysaccharide (LPS) before and after the infection and 20 ng mL$^{-1}$ mrIL-4 (PeproTech) 4 h postinfection. CSR efficiency in the infected (GFP$^+$) subpopulations was measured by flow cytometry using biotinylated anti-IgG1 (BD) followed by anti-biotin-allophyocyanin (Miltenyi Biotec) and propidium iodide to exclude dead cells. CSR in CH12 B cells was induced with CIT [1 µg mL$^{-1}$ rat-anti-CD40 (clone 1C10, eBioscience), 10 ng mL$^{-1}$ interleukin (IL)-4 and 1 ng mL$^{-1}$ transforming growth factor-β1 (R&D Systems)]. The proportion of IgA+ cells was measured 3 days later by flow cytometry using anti-IgA conjugated with R-phycoerythrin (SouthernBiotech). Gating strategies for CSR are shown in Supplementary Fig. 8b, c. SHM at Sµ was analysed in $Aicda^{-/-}$ $Ung^{-/-}$ mouse B cells transduced twice with pMX-AID variant-ires-GFP in order to get nearly 100% infection efficiency. Infected cells were cultured for 4 days with 10 µg mL$^{-1}$ LPS and 25 ng mL$^{-1}$ IL-4 before enriching for live cells using OptiPrep (Sigma) and purifying genomic DNA with DirectPCR lysis reagent (Viagen Biotech). To amplify mutations at the Sµ region, we amplified a 1749 bp fragment starting at the TSS in experiments using full-length AID or 607 bp fragment overlapping the S-region in experiments with AIDΔE5, using KOD1 DNA polymerase. SHM at the IgV in DT40 cells was analysed in cells complemented with AIDΔE5 variants and cultured for 3 days (when IgM-loss levels were not saturated, see Fig. 8e). GFP+ cells were sorted but no phenotypic selection for IgM was used to be able to calculate actual mutation frequencies. Genomic DNA was immediately purified using DirectPCR and the IgV region was amplified[36]. Oligonucleotide sequences are in Supplementary Table 2. Amplicons were cloned into pGEMT-easy (Promega), and individual clones were sequenced at Macrogen (Seoul, Korea). The distribution and frequency of mutations were computed as described[36]. Briefly, sequences were aligned using Sequencher (Gene Codes Corp.) and trimmed to remove vector and primer sequences, and electropherograms were manually inspected to confirm bona fide mutations before compiling the total and per sequence mutation numbers.

**Competitive growth assays.** The effect on cell fitness was analysed by using AID-deficient CH12 B cells complemented with AID variants-ires-GFP or empty vector (GFP) by retroviral transduction. Depending on the transduction efficiency, the population of infected cells was sometimes kept as is or mixed with untransduced (GFP−) cell. Cells were maintained in culture with normal medium monitoring the GFP+ to GFP− cell ratio over time by flow cytometry immediately after transduction. Gating strategy is shown in Supplementary Fig. 8d. In some experiments, cells 1 nM DMSO or Did B were added to the medium. The GFP+ to GFP− cell ratio over time was monitored by flow cytometry. Gating strategy is shown in Supplementary Fig. 8d.

**Deaminase activity and DNA-binding assays.** *E. coli* mutation assays were performed using the $\Delta ung$ BW310 strain transformed with AID variants subcloned as BamHI-XhoI fragments into pTrcHisA (Invitrogen). His-AID fusions were expressed by 1 mM Isopropyl β-D-1-thiogalactopyranoside induction for 16 h at 37 °C and plated on rifampicin or ampicillin LB plates. Mutation frequencies were calculated as the median number of colony-forming units that survived rifampicin selection per $10^9$ ampicillin-resistant cells from 2 to 5 experiments with 5 independent cultures per construct. For biochemical assays, EcoRI fragments encoding the open reading frame of each AID were cloned into pGEX-5×-3 (GE Healthcare) to generate and purify GST-AID as described[28]. For each mutant and wt AID, 2–4 independent preparations were purified and tested. An end-labelled bubble substrate containing a 7-nt-long single-stranded region with the motif TGC, previously

described to be an optimal AID substrate, was used in activity assays and electrophoretic mobility shift assays (EMSAs)[60]. For alkaline cleavage, 0.03–4 nM substrate was incubated with 0.1 µg AID, followed by addition of UDG, NaOH and heat, and electrophoresis on denaturing urea gels, as described[28,60]. For EMSA, 0.015–5 nM substrate was incubated with 0.1 µg of GST-AID in binding buffer (50 mM Tris, pH 7.5, 2 µM MgCl, 50 mM NaCl and 1 mM DTT) in a final volume of 10 µl for 60 min at 37 °C, followed by UV cross-linking as previously described[60]. Samples were electrophoresed at 4 °C on an 8% acrylamide native gel. Alkaline cleavage and EMSA gels were visualized using a PhosphorImager (Bio-Rad). Densitometry was performed using the Quantity One 1-D Analysis Software (Bio-Rad). Data were graphed using GraphPad Prism to derive initial deamination velocity and $K_d$ values.

**AID shuttling and nuclear wash protocol.** Retrovirally complemented AID-deficient CH12 cells were treated for 2 h with DMSO, 10 ng mL$^{-1}$ LMB (a CRM1 inhibitor), 100 nM Did B (an EEF1A inhibitor) or both drugs combined before harvesting. Cells were washed with phosphate-buffered saline (PBS), then plated on poly-L-lysine-coated coverslips and fixed in 3.7% (w vol$^{-1}$) formaldehyde for 10 min and then washed 3× in PBS. The nuclear wash protocol was adapted from ref. [38]. Briefly, CH12 cells were plated on poly-L lysine coverslips and washed 1× with CSK buffer (10 mM PIPES, 300 mM sucrose, 200 mM NaCl, 3 mM MgCl₂, 1 mM EDTA and 1 × fresh complete protease inhibitor (CPI, Roche)). Cells were then either fixed directly in formaldehyde (whole cell) or washed to remove cytoplasm and loosely held nuclear proteins. Washing was done by sequentially incubating the coverslips on ice in: CSK buffer for 1 min, CSK+ 0.1% triton X-100 for 1 min, CSK +0.5 % triton X-100 for 20 min. For nuclease treatments, the last wash was 10 min on ice, then 10 min at 37 °C in CSK buffer containing 10 mg mL$^{-1}$ RNase or 10 mg mL$^{-1}$ DNase. After washes, cells were fixed in formaldehyde. For all IF, cells were permeabilized and blocked for 1 h in blocking solution (PBS, 0.5% (v/v) Triton-X100, 1 mg mL$^{-1}$ bovine serum albumin (BSA), 5% (v/v) goat serum). For anti-LaminB IF, blocking buffer was 5% BSA to avoid cross-reactivity of anti-goat secondary. Cells were then incubated overnight at 4 °C with primary antibodies in blocking solution, followed by 3× washes with PBS+0.01% Triton X-100 (PBS-T), then a 1 h incubation with secondary antibodies in blocking solution and 3× PBS-T washes. After nuclear staining with 4,6-diamidino-2-phenylindole (Dapi; 300 nM in PBS), coverslips were washed with ddH₂O and mounted on slides using Lerner Aqua-Mount (Thermo Scientific).

**LacR-LacO recruitment.** U2OS cells with a 256 copy *lacO* array[40] (a gift from Dr. A. Orthwein, McGill University) were plated on coverslips and co-transfected with mCherry-LacR-NLS or mCherry-LacR-NLS-SPT5 along with GFP-tagged AID variants or APOBEC1 using TransIT-LT1 transfection reagent (Mirus). Thirty-to-40 h post-transfection, cells were fixed in 3.7% (w/v) formaldehyde for 10 min, then washed 3×, stained with Dapi, washed and mounted as above.

**Microscopy.** Images were acquired at room temperature using ZEN 2010 on a Zeiss LSM 700 confocal microscope with excitation lasers at 405 nM (Dapi), 488 nM (GFP), 543 nM (Alexa546 and DyLight550) and 633 nM (Alexa680), using either 40×/1.3 or 63×/1.4 oil immersion objectives, and collected with a Hamamatsu PMT. Settings for nuclear wash: endogenous AID, whole cell (laser power 5, gain 550), nuclear wash (laser power 20, gain 650); overexpressed AID, whole cell (laser power 5, gain 475), nuclear wash (laser power 5, gain 625). Subcellular localization was scored in Volocity (Perkin Elmer). Masks were made for each individual cell for both nuclear and total IF signal. The proportion of nuclear signal was calculated as the ratio of nuclear signal/total signal×100. For nuclear washes, whole-cell IF signal was measured from a mask generated by GFP signal, whereas nuclear signal was measured from a mask generated by Dapi signal. For γH2AX foci quantification, foci were counted in each transduced, GFP+ cell, using Volocity spot counting within a nuclear mask generated by Dapi signal. For each experiment, multiple fields were analysed, excluding cells showing saturated signal, abnormal DNA structure or mitotic figures. For making figures, images were transferred to Photoshop for cropping and adjusting contrast throughout the whole image when necessary to enhance visibility.

**Chromatin fractionation.** Chromatin fractionation was adapted from ref. [39]. Briefly, ~50 × 10⁶ CH12 cells were collected and washed 1× with ice-cold PBS prior to re-suspension in 1 mL of Lysis buffer (10 mM HEPES pH 7.9, 10 mM KCl, 1.5 mM MgCl₂, 0.34 M Sucrose, 10 % glycerol, 0.1 % Triton X-100, 1 mM DTT, 1 × CPI). Cells were lysed for 8 min on ice and then centrifuged at 1300 × g for 4 min. The supernatant was kept as the cytoplasm fraction. The pellet was resuspended in 200 µL nuclear resuspension buffer (10 mM Tris pH 8, 3 mM CaCl₂, 1 mM Mg Acetate, 0.34 M Sucrose, 0.5% NP-40, 1 mM DTT, 1× CPI). Nuclei were then layered onto a sucrose cushion (10 mM Tris pH 8, 2 M Sucrose, 5 mM Mg Acetate, 0.1 mM EDTA, 1 mM DTT), and centrifuged at 20,000 RPM for 15 min. The nuclear pellet was resuspended in 500 µL of nuclear resuspension buffer without NP-40 and then centrifuged at 100 × g for 10 min. For nuclear–cytoplasmic fractionation, the protocol was stopped here and extracts analysed by WB. For further fractionation, nuclei were washed 1× with nuclear wash buffer (10 mM Tris pH 7.4, 2 mM MgCl2, 1× CPI), then resuspended in 400 µL of nuclear wash buffer and

placed at 37 °C for 5 min. CaCl$_2$ was added to 1 mM and DNA digested by adding Mircococcal nuclease (New England Biolabs) to 9.6 U mL$^{-1}$ for 10 min. Digestion was stopped by adding EGTA to 2 mM final. An aliquot was saved as the total nuclear fraction. Nuclei were pelleted for 10 min at 100 × *g*, and the supernatant was saved as the MNase fraction. Nuclei were then re-suspended in 700 μL of 150 mM extraction buffer (10 mM Tris pH 7.4, 140 mM NaCl, 1 mM MgCL$_2$, 2 mM EGTA, 0.1 % Triton X-100, 1× CPI) and incubated for 2 h on a rocker at 4 °C. After centrifugation for 10 min at 100 × *g*, the supernatant was saved as the 150 mM fraction. Nuclei were then re-suspended in 700 μL of extraction buffer, with 590 mM NaCl, and incubated overnight on a rocker at 4 °C. After centrifugation for 10 min at 500 × *g*, the supernatant was saved as the 600 mM fraction. Nuclei were finally resuspended in 10 mM Tris pH 7.4, 200 mM NaCl, 1 mM EDTA, 1× CPI and kept as the pellet fraction. DNA was purified using a PCR Purification Kit (BioBasic, Cat.# BS614) and then run on an agarose gel in order to confirm efficient digestion and DNA extraction. Protein fractions were analysed by sodium dodecyl sulfate-polyacrylamide gel electrophoresis (SDS-PAGE) and WB.

**Chromatin immunoprecipitation**. For ChIP in primary B cells, naive *Aicda*$^{-/-}$ mouse B cells were stimulated with 12 μg mL$^{-1}$ LPS and 50 ng mL$^{-1}$ IL-4 and retrovirally transduced 24 h later. Cells were harvested at 18–20 h postinfection when GFP+ cell proportions were 30–50%. For ChIP in DT40 *Aicda*$^{-/-}$ ΔΨVλ cells, repeated retroviral transduction achieved ~75–90% GFP+ cells, and cells were expanded for 48 h. ChIP procedures have been described in detail[58]. Briefly, cells fixed with 1% formaldehyde for 30 min, were lysed in RIPA buffer and sonicated to generate DNA fragments 500 bp. Lysate fractions (2 μg μL$^{-1}$) of 0.5 mg (for endogenous AID) or 1.25 mg (for endogenous SPT6) were precleared with G protein-Sepharose slurry before incubating overnight with 2–5 μg anti-AID antibody (328.8b, Active Motif) at 4 °C. In ChIPs from DT40 extract, Protein G magnetic microbeads were used and IP purified by μMACS columns (Miltenyi Biotech). DNA was purified and used as template in real-time PCR reactions containing 1× SYBR Green Mix (Applied Biosystems), 1/10 fraction ChIP-enriched DNA and 100 nM primers (see Supplementary Table 2 for primer sequences). Plates were read in an Applied Biosystems StepOnePlus instrument. Standard curves with different amounts of the input extracts were run in each plate for each individual amplicon and used to calculate input %. The input % of the IgG control immunoprecipitation was subtracted from each sample to calculate the values reported.

**Co-immunoprecipitation**. AID-deficient CH12 cells were reconstituted with GFP-tagged AID variants then lysed as in ref. [17]. Briefly, 50 × 10$^6$ cells were resuspended in 400 μL of low salt buffer (20 mM HEPES pH 7.5, 10 mM KCl, 1 mM MgCl$_2$, 10% glycerol, 1% NP40, 1× CPI) containing Benzonase nuclease (Sigma) and were sonicated 2 × 10 s at 50% amplitude on ice. Samples were then incubated 30 min and clarified at max. speed for 10 min. Supernatant was collected and mixed with 250 μL of high salt buffer (low salt buffer containing 400 mM NaCl). The pellet was then re-extracted with low salt buffer (no Benzonase) as above, and the supernatant was again supplemented with high salt buffer after clarification. The pellet was finally resuspended in 400 μL of high salt buffer and subjected to same extraction. All three supernatants were pooled (final NaCl concentration of 200 mM) and clarified one last time, before GFP immunoprecipitation was carried out using the μMACS GFP Isolation Kit according to the manufacturer's instructions (Miltenyi Biotec, Cat.# 130-091-125). Elution and total lysate were analysed by SDS-PAGE and WB.

**BioID and mass spectrometry**. BioID samples were processed as described elsewhere[61], with modifications. Briefly, for each construct, 18 × 10$^6$ *Aicda*$^{-/-}$ mouse B cells were pre-cultured for 48 h with 0.5 μg mL$^{-1}$ anti-CD180 (BD Biosciences). Cells were then infected twice over consecutive days with pMX-AID variants-BirA*-Ires-GFP or pMX-A2-BirA*-Ires-GFP retrovirus in the presence of 5 μg mL$^{-1}$ LPS. After the second infection, media was supplemented with 5 μg mL$^{-1}$ LPS+25 ng mL$^{-1}$ IL-4. The next day, the media was supplemented to 50 μM biotin (Sigma). Cells were harvested 24 h later (~40–50 × 10$^6$ cells), washed 3× with PBS, then lysed in 1.5 mL of RIPA buffer and sonicated 30 s at 30% amplitude (3 × 10 s bursts with a 2 s break in between). Benzonase (250 U, EMD Millipore) was added to the lysates during centrifuging, 30 min at 16,000 × *g*, 4 °C. Forty μL aliquots of supernatant were kept to monitor expression and biotinylation, and the remaining lysate was incubated with 70 μL of pre-washed streptavidin-sepharose beads (GE Healthcare) for 3 h on a rotator at 4 °C. Beads were then washed with 1 mL of RIPA buffer, transferred to a new tube and washed again 2× with 1 mL of RIPA buffer and then 3× with 1 mL of 50 mM Ammonium Bicarbonate (ABC) (Biobasic). Beads were then resuspended in 100 μL of ABC with 1 μg of trypsin (Sigma) and incubated overnight at 37 °C with rotation. The following day, 1 μg of trypsin was added for a further 2 h digestion. Samples were centrifuged 1 min at 2000 RPM, and the supernatant was transferred to a new tube. Beads were rinsed twice with 100 μL of water, and all supernatants were pooled and adjusted to 5% formic acid. Samples were then centrifuged for 10 min at 16,000 × *g* for clarification. Trypsin-digested peptides in the supernatant were dried in a SpeedVac (Eppendorf) for 3 h at 30 °C. Samples were resuspended in 15 μL of 5% formic acid and kept at −80 °C for mass spectrometric analysis. Samples were injected into an Orbitrap Fusion (Thermo Fisher). Protein identification and analysis was carried out as described elsewhere[61]. Briefly, RAW files were converted to.mzXML in Proteowizard[62]. Peptide search and

identification was processed using Human RefSeq Version 57 and the iProphet pipeline[63] integrated in ProHits[64].

**Statistical analyses**. We used four different methods to identify wt AID–BioID interactions that were significantly and consistently reduced in both R-mutants tested. Comparative results for the four methods are provided in Supplementary Table 1. All methods were implemented through ad-hoc scripts in R version 3.3.2 that are available upon request. Tables were handled in R using the data.table package. Method 1 used fold-enrichment ('Fold'), calculated using mean spectral counts (s.c.). First, s.c. were normalized in all samples to their corresponding BirA* s.c. levels. Interactors with at least 2.5-fold enrichment in wt AID over the R mutants or vice versa were considered differential interactors. To eliminate interactions that were not AID-specific, only interactions with at least five-fold enrichment over APOBEC2 were considered. Method 2 calculated a *Z*-score using log-transformed fold-enrichment values ('Normz'). Method 3 calculated local *Z*-scores using a sliding window of 10% of the data points around the candidate in an ratio-intensity R-I plot ('Maz')[65]. In methods 2 and 3, hits with global or local, respectively, *Z*-scores ≥2 (i.e., 2 s.d. from the mean) were considered as differential interactors. Methods 1–3 assume normal distribution of the data. Method 4 used DESeq2 v.1.14.1[66], an R package that uses generalized linear models to fit a negative binomial distribution to the data ('Deseq'). The R-mutants were defined as reference and default parameters from the package were used to carry out the analysis and calculate differential interactions for AID. Preys with multiple-testing adjusted *P*-values (Benjamini–Hochberg procedure) <0.1 were considered as significant. All figures were plotted using ggplot2 package in R. Dot plots were generated using the Prohits-Viz web tool[67].

Where indicated, for pairwise comparisons of treatments or conditions, differences were evaluated by unpaired, two tailed *t*-test with alpha = 0.05 using Prism (GraphPad).

**Code availability**. R scripts used for statistical analyses of BioID data are available from the corresponding author upon reasonable request.

**Data availability**. Sequences of mouse Sμ and DT40 IgV regions have been deposited in GenBank as follows: For Sμ region in Fig. 8g, AIDΔE5 MG904385-MG904404, AIDΔE5 R171Y MG904405-MG904427, AIDΔE5 R174E MG904428-MG904449, AIDΔE5 R178D MG904450-MG904469. For DT40 IgV region in Fig. 8f, AIDΔE5 MG904470-MG904484, AIDΔE5 R171Y MG904485-MG904522, AIDΔE5 R174E MG904523-MG904557, AIDΔE5 R178D MG904558-MG904592. For Sμ region in Fig. 3a, AID MG904593-MG904622, AID R171Y MG904623-MG904654, AID R178D MG904655-MG904685. Mass spectrometric data have been deposited in MassIVE under ID MSV000081963. All other data generated or analysed during this study are included in this published article and its supplementary information files.

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

## Acknowledgements

We thank Dr. N. Francis for critical reading and Dr. F. Robert, Dr. A. Orthwein and Dr. A. Zahn for discussions. We thank Jean-Felix Côté for technical help and the assistance of D. Fillion with microscopy, E. Massicotte, J. Lord, J. Leconte and P. St-Onge with flow cytometry, D. Faubert with mass spectrometry and M. Cawthorne and E. Thivierge with animal care. We thank Dr. M. Neuberger, Dr. A. Orthwein, Dr. D. Durocher, Dr. R. Pavri and Dr. T. Honjo for sharing reagents. This work was funded by grants from the Cancer Research Society (to J.M.D.N.) and Canadian institutes of Health research (MOP130535

to J.M.D.N., MOP111132 to M.L.), CCSRI innovation grant 702145 (to M.L.), NIH R01GM121595 (to R.E.V.) and a Discovery grant from The Natural Sciences and Engineering Research Council of Canada (to J.-F.C.). S.P.M. and H.B. were supported by doctoral training awards from the Fonds de Recherche du Québec-Santé. S.P.M. and L.C. L. were supported by doctoral fellowships from the Cole Foundation. J.-F.C. holds the Transat Chair in Breast Cancer Research and is supported by an FRQS Senior investigator career award. J.M.D.N. holds the Canada Research Chair in Genetic Diversity.

## Author contributions

Conceptualization, S.P.M., L.C.L. and J.M.D.N.; methodology, S.P.M. and L.C.L.; formal analysis, P.G.S.; investigation, S.P.M., L.C.L., A.K.E., H.F., A.-M.P., J.C.G., H.B., G.E.S. and R.E.V.; data curation, P.G.S and H.B.; writing—original draft, S.P.M. and J.M.D.N.; writing—reviewing and editing, S.P.M., L.C.L., P.G.S., H.B., J.-F.C., M.L., R.E.V. and J.M. D.N.; funding acquisition, J.M.D.N.; supervision, J.-F.C., M.L., R.E.V. and J.M.D.N.

## Additional information

**Competing interests:** The authors declare no competing interests.

