## [Peer Review File · Nature Communications]

Reviewers' comments:

Reviewer #1 (VDJ, RAG, SHM/CSR)(Remarks to the Author):

This manuscript investigates the phenotype of RR domain mutants of AID. The data reveal that mutation of any one of 3 Arg residues in alpha-helix 6 (the RR domain) yields an interesting phenotype of poor SHM and CSR but normal or nearly normal catalytic activity, cell trafficking, and general association with chromatin. Instead, these mutants fail to associate normally with Smu, while being present at normal levels at the Imu promoter, and also fail to be in proximity with a number of interesting transcription and chromatin-associated factors, as revealed by a biotinylation-proximity screen. The results lead to an interesting, novel concept of a "licensing" step required for AID to associate with elongating Pol II.

The manuscript is for the most part extremely solid: the phenotypes of the mutants are extensively investigated (one of the most extensive and rigorous such studies in the field); the experiments are well controlled (with two important exceptions, noted below); the data are nicely presented and appropriately interpreted; and the findings are both novel and important. In particular, the use of AID-AID fusion proteins for complementation studies and the AID-BirA* proximity screen are firsts for the field and provide important new observations, and also set the stage for extensive future studies. The resulting model is provocative and will help guide future work in the field. As detailed below, there are two control experiments that need to be added, and it is important for the authors not to oversell their model; as detailed in my last comment, they are advised to discuss the caveats and limitations of the study more fully so that readers are not misled. With these changes/additions, I would strongly support publication in Nature Communications.

Specific comments.

1. Page 7, line 23: "kinetics" is not the right word, since this is an affinity measurement.

2. For the most part, the experiments are very well controlled. The exception appears to be Figure 5b, where recruitment of AID by mCherry-LacR-Spt5 is assessed. The authors conclude that the recruitment is due to the presence of Spt5, but do not do the key control of testing recruitment by mCherry-LacR. Without this, the main conclusion of the panel is not warranted. In particular, the conclusion on page 18, lines 11-12 "with Spt5 being necessary and sufficient to tether AID to chromatin" is not warranted. Even if the proper control were added and showed that mCherry-LacR failed to recruit AID, it is hard to understand why the authors would conclude that Spt5 is necessary to tether AID to chromatin; the data with RNase treatment undermines this claim. In fact, in the very next sentence, the authors state that there is a pool of AID that is not Spt5-dependent; hence, Spt5 is not necessary to tether AID to chromatin. The authors should also re-think their claim that it is sufficient, since even in the experiment of Fig. 5b, they do not know what other factors might be recruited by mCherry-LacR-Spt5. The authors are encouraged to dispense with the "necessary and sufficient" terminology and state their conclusions more precisely. The same problem appears on page 25, line 12.

3. A second example of a missing control comes from Figure 6a. The authors wish to conclude that the E58A domain is contributing RR domain function in trans, but they do not do the key experiment with a fusion protein in which the E58A domain is mutated in an R residue of the RR domain. This is important to support their conclusion.

4. Page 18, line 4: "a cause effect" is not correct usage.

5. Figure legend for Fig. 6d: what is "sc"?

6. The discussion and the model of Fig. 7f are interesting and provocative, but the authors are encouraged to address the limitations of their data and model more fully. For example, on p26, lines 7-9: "In fact, our results imply that in physiological conditions AID is recruited to the TSS and must progress with transcription elongation until the polymerase becomes stalled, rather than being directly recruited to stalled polymerases." The evidence for AID progression with elongating Pol II is largely derived from the ChIP experiments of Fig. 7a, combined with indirect inferences derived from the BirA* interactome data. The ChIP data pertain exclusively to CSR, and there are multiple, plausible explanations for reduced AID association at Smu in the RR mutants (especially given the findings recently reported in ref 54) ; reduced association with elongating Pol II is only one of them. Further, it is not clear to what extent the interactome data reflect events during SHM versus CSR. Hence, the degree to which the model explains poor SHM by RR-domain mutants is unclear. I do not have a major problem with the sentence quoted above, but I feel that limitations in the data (as they pertain to the model and to relevance to SHM) are not addressed sufficiently.

Reviewer #2 (AID, B VDJ/SHM)(Remarks to the Author):

In their manuscript entitled "A licensing step links AID to transcription elongation for B cell mutagenesis", the authors mainly focused on three arginine residues in AID's alpha-6 motif, and found that these R-mutants mutate E.coli with the same efficiency as wildtype AID but fail to induce CSR and SHM (Fig 1). Then the authors made efforts to elucidate the underlying mechanism. On the one hand, they showed that the R-mutants have normal DNA binding kinetics (Fig 1), unchanged ssDNA deamination activity (Fig 2), normal nucleocytoplasmic shuttling (Fig 2), regular association with nuclear complexes (Fig 4), and similar dependence on Spt5/RNAPII for chromatin association (Fig 5). On the other hand, the authors showed that the R-mutants exhibit reduced association with the Igh Su locus (Fig 3 and 7) and with a series of proteins, including Spt6 (Fig 6). From these observations, the authors conclude that R-mutants lack association with the transcription elongation machinery and lack a licensing mechanism for AID's mutagenic activity.

Although the authors convincingly showed that these specific replacements of arginine residues abrogate SHM and CSR without affecting AID's enzymatic activity, overall the manuscript would benefit if attention were paid to the following.

Major points:

1) The observations described in this manuscript can be explained by recently published AID structure and in vitro data. As acknowledged by the authors in the Discussion part, it has been shown recently by AID structure why the mutations on RR domain affect AID deamination activity. In Qi's paper (Qi et al. Mol Cell. 67, 1-13; Fig 5), it is shown that one AID molecule interacts with two single-stranded DNA (ssDNA) overhangs: one is the substrate for deamination, while the other is for assistance to enhance affinity. This "assistant patch" is exactly the RR domain mentioned in the authors' manuscript. Moreover, Qi and his colleagues also showed that mutations on R171, R174, R177, and R178 specifically compromised AID deamination activity on substrates with two ssDNA overhangs, but not significantly altered on substrate with only one ssDNA overhang. These structure and in vitro data in Qi's paper perfectly explained the authors' observation in this manuscript that these R-mutants abolish activity for CSR but still maintain enzymatic activity. However, the authors' explanation seems less

convincing and the novelty is limited.

2) The conclusions in the last part of the manuscript need some additional strengthening. The authors conclude that "AID R-mutants fail to progress with elongating RNAPII", and it is based on only two facts: 1) AID R-mutants interacts less abundantly with Spt6, which may participate in transcription elongation; and 2) AID ChIP qPCR data showed that the R-mutants were depleted from Su region but still enriched at the TSS. This conclusion needs to be further strengthened, for example, by experiments to show that AID loading is only reduced around Su region but not at TSS in Spt6-deficient cells. Moreover, genome-wide ChIP Sequencing needs to be performed so that other switch regions or off-target sites could be examined.

3) The proposed licensing mechanism lacks evidence. The authors concluded that the licensing steps could be bypassed by extending AID's residence in the nucleus (Fig 7d and 7e). However, as shown in Fig 3b, R-mutants did not totally abolish their deamination activity. Therefore, accumulated nuclear residence leading to high mutation level is not very surprising. Moreover, deletion of AID's C-terminal nuclear exporting signal would affect AID's stability. Therefore, the authors need to show that $\Delta E5$, $\Delta E5$ -R171Y, $\Delta E5$ -R174E, and $\Delta E5$ -R178D have similar protein stability in Fig 7d and 7e. In the Discussion part, the authors proposed "high nuclear AID levels can bypass the licensing step, possibly by direct recruitment to stalled polymerases." If this were true, deletion of AID's C-terminal nuclear would lead to higher interaction with Spt5, but the authors did not provide these data.

Additional points:

1) In the arginine mutation experiment, R174 is mutated to E; while R177 to A, and R178 to D. What is the reason for these different mutations? The authors stated, "AID R177A maintained all three activities". R is positively charged; D/E are negatively charged; and A is non-polar. Therefore, if the positively charged R177 were mutated to negatively charged D or E, maybe the CSR or SHM would be affected.

2) It is a little confusing that in Fig 6g R-mutants co-immunoprecipitated Spt6, while in Fig 6h R-mutants could not interact with Spt6. The author reasoned it as "the power of BioID to detect functional defects in live cells". It would be appropriate to explain more here or to give a reference.

3) It is very interesting that a series of interaction proteins with AID were identified. A lot of them have not been reported before. Further investigation would facilitate the understanding of the functional deficiency of R-mutants.

4) Fig 3a and Fig 7a are repeated data, which could be combined.

5) In page 12, "Fig. 3f-h" needs to be changed to "Fig. 3e-g".

Reviewer #3 (AID, VDJ, RAG, DNA repair) (Remarks to the Author):

In this study, Methot and colleagues showed that three arginine residues in AID $\alpha 6$ domain are critical for both CSR and SHM in B cells, but are dispensable for cytidine deamination in *E. coli* or in vitro. The arginine mutations impair the interaction of AID with transcription elongation and correspondingly reduce the recruitment of AID to S μ . Based on these data, the authors conclude that the arginine

residues play an important role in coupling transcription elongation with AID function, and refer to the coupling mechanism as a “licensing step” in AID-mediated mutagenesis. It has been known that AID function is closely linked to transcription, but the underlying mechanism is not fully understood. This study represents a significant step forward towards solving this important question. Overall, the data on the characterization of AID mutants in various assays are convincing and the study is appropriate for Nature Communications.. The authors could improve the study by addressing to the following comments:

- 1) A central piece of evidence for the licensing model is the observation that AID R-mutants occupy the promoter region, but not the body of S μ . To generalize the model, the authors may wish to extend this analysis to an additional physiological target of AID such as another S regions or Ig variable region if the latter is experimentally feasible. In any case, it would be helpful if they could include a highly transcribed non-target gene for as a control for the specificity of AID ChIP.
- 2) The authors argue that the licensing mechanism is enforced by limiting nuclear levels of AID. To strengthen this claim, the authors might use the ChIP assay to demonstrate that E5 deletion can rescue the recruitment defect of R-mutants.
- 3) In figure 2, the authors should consider complementing IF assay with Western analysis of AID protein levels in nuclear and cytoplasmic fractions.
- 4) R-mutations decrease the interaction between AID and Spt6. However, the authors provided no evidence to demonstrate that the loss of this interaction is in fact responsible for the CSR or SHM defect. This point should be mentioned.
- 5) As shown in Figure 1c and d, the expression level of R174E protein is lower than WT protein. Furthermore, as judged by visual inspection of the image in Figure 2a, R174E protein remains largely cytoplasmic even in the presence of LMB and DidB. Thus, the authors should note the possibility that the CSR and SHM defect associated with R174E mutation may be attributed to reduced expression and nuclear accumulation.

Point-by-point response to Reviewers

Reviewer #1 (VDJ, RAG, SHM/CSR)(Remarks to the Author):

This manuscript investigates the phenotype of RR domain mutants of AID. The data reveal that mutation of any one of 3 Arg residues in alpha-helix 6 (the RR domain) yields an interesting phenotype of poor SHM and CSR but normal or nearly normal catalytic activity, cell trafficking, and general association with chromatin. Instead, these mutants fail to associate normally with Smu, while being present at normal levels at the I_{mu} promoter, and also fail to be in proximity with a number of interesting transcription and chromatin-associated factors, as revealed by a biotinylation-proximity screen. The results lead to an interesting, novel concept of a "licensing" step required for AID to associate with elongating Pol II.

The manuscript is for the most part extremely solid: the phenotypes of the mutants are extensively investigated (one of the most extensive and rigorous such studies in the field); the experiments are well controlled (with two important exceptions, noted below); the data are nicely presented and appropriately interpreted; and the findings are both novel and important. In particular, the use of AID-AID fusion proteins for complementation studies and the AID-BirA* proximity screen are firsts for the field and provide important new observations, and also set the stage for extensive future studies. The resulting model is provocative and will help guide future work in the field. As detailed below, there are two control experiments that need to be added, and it is important for the authors not to oversell their model; as detailed in my last comment, they are advised to discuss the caveats and limitations of the study more fully so that readers are not misled. With these changes/additions, I would strongly support publication in Nature Communications.

We thank the Reviewer for the very positive comments and constructive criticism. We have addressed all the points raised, as detailed below.

Specific comments.

1. Page 7, line 23: "kinetics" is not the right word, since this is an affinity measurement.

Indeed. We have replaced "kinetics" for "affinity".

2. For the most part, the experiments are very well controlled. The exception appears to be Figure 5b, where recruitment of AID by mCherry-LacR-Spt5 is assessed. The authors conclude that the recruitment is due to the presence of Spt5, but do not do the key control of testing recruitment by mCherry-LacR. Without this, the main conclusion of the panel is not warranted. In particular, the conclusion on page 18, lines 11-12 "with Spt5 being necessary and sufficient to tether AID to chromatin" is not warranted. Even if the proper control were added and showed that mCherry-LacR failed to recruit AID, it is hard to understand why the authors would conclude that Spt5 is necessary to tether AID to chromatin; the data with RNase treatment undermines this claim. In fact, in the very next sentence, the authors state that there is a pool of AID that is not Spt5-dependent; hence, Spt5 is not necessary to tether AID to chromatin. The authors should also re-think their claim that it is sufficient, since even in the experiment of Fig. 5b, they do not know what other factors might be recruited by mCherry-LacR-Spt5. The authors are encouraged to dispense with the "necessary and sufficient" terminology and state their conclusions more precisely. The same problem appears on page 25, line 12.

Thank you for raising these points.

Firstly, we have now included the requested mCherry-LacR control in **Figure 5b**. This control was part of our original experiments but we left it out from the figure for simplicity, as we felt the Apobec1 control was sufficient. However, we agree with the Reviewer that this control better demonstrates the role of Spt5 in the recruitment of AID to chromatin.

Secondly, we also agree that we were not precise or clear enough in our interpretation and discussion of the role of Spt5 for AID chromatin association, which we have modified. We interpret that Spt5 plays a necessary role in maintaining the association of AID to chromatin because Spt5 depletion by shRNA leads to AID depletion from the chromatin (Figs 5a). However, we realize now that by describing this activity as "tethering" we were implying a direct effect, which we do not demonstrate. It is likely that other factors, as shown by the RNase treatment are also necessary for retaining AID. For the same reason qualifying Spt5 as sufficient to recruit AID was misleading, as we cannot rule out that there are intermediate factors mediating the recruitment of AID to the Spt5 foci. We now interpret the mCherry-LacR experiment (Fig 5b) to show that forcing an accumulation of Spt5 at the chromatin is sufficient to induce the recruitment AID, which extend our understanding of the AID-Spt5 interplay compared to previous work, but mention that this effect could be indirect. We have modified the text in the corresponding section of Results and substantially changed the discussion to reflect these facts.

3. A second example of a missing control comes from Figure 6a. The authors wish to conclude that the E58A domain is contributing RR domain function in trans, but they do not do the key experiment with a fusion protein in which the E58A domain is mutated in an R residue of the RR domain. This is important to support their conclusion.

We thank the reviewer for suggesting this control, which we had overlooked. We have now produced this control for each AID^{Rmut} --- AID^{E58A} fusion protein (i.e. the corresponding AID^{Rmut} --- $AID^{E58A+Rmut}$). Each of them loses CSR activity, demonstrating that the enzymatically dead AID^{E58A} is indeed mediating the recovery in CSR activity through the RR domain function. Importantly, we have also verified that the new fusions retain equal catalytic activity than the parental fusion, as judged by *E coli* mutation assays. This data has been integrated into **Figure 6a**.

4. Page 18, line 4: "a cause effect" is not correct usage.

This has been corrected.

5. Figure legend for Fig. 6d: what is "sc"?

We apologize for the omission; sc is "spectral counts". This is now defined in the legend and text.

6. The discussion and the model of Fig. 7f are interesting and provocative, but the authors are encouraged to address the limitations of their data and model more fully. For example, on p26, lines 7-9: "In fact, our results imply that in physiological conditions AID is recruited to the TSS and must progress with transcription elongation until the polymerase becomes stalled, rather than being directly recruited to stalled polymerases." The evidence for AID progression with elongating Pol II is

largely derived from the ChIP experiments of Fig. 7a, combined with indirect inferences derived from the BirA* interactome data. The ChIP data pertain exclusively to CSR, and there are multiple, plausible explanations for reduced AID association at Smu in the RR mutants (especially given the findings recently reported in ref 54) ; reduced association with elongating Pol II is only one of them. Further, it is not clear to what extent the interactome data reflect events during SHM versus CSR. Hence, the degree to which the model explains poor SHM by RR-domain mutants is unclear. I do not have a major problem with the sentence quoted above, but I feel that limitations in the data (as they pertain to the model and to relevance to SHM) are not addressed sufficiently.

This comment raises two fair points that we have addressed.

One point is the general validity of our model. In our original manuscript, we extrapolated from qChIP results at the S μ region to propose that licensing permitted AID to occupy the gene bodies after loading at the promoter. To address this limitation, we have performed qChIP analyses of additional genes: the acceptor CSR S-region (S γ 1) and a known AID off-target gene (Il4 α) in primary mouse B cells, as well as the IgV region in DT40 cells undergoing SHM. In every case we found that the R-mutants associate normally to the promoter but are depleted from the gene body. These results are fully consistent with our model generally applying to CSR and SHM, and to at least a subset of off-targets exemplified by Il4 α . Extending it to all off-targets will require genome wide studies, as we acknowledge in the Discussion.

As suggested we have also made a more nuanced Discussion of our model. The reviewer is especially concerned that the reduced association of the R-mutants with the S μ could be explained by the recent report of Qiao et al (Mol Cell 67, 361-73, 2017). This work proposed an assistant DNA binding patch in AID, formed by the same arginine residues in α -helix 6 that we studied and defined as the RR domain. Qiao et al showed that mutations in those residues impaired the ability of purified AID to deaminate structured DNA in vitro, forked and more so G4 DNA oligonucleotides. However, the effect on G4 DNA deamination of each mutation was different, and never complete, while the defect in CSR was virtually complete. Although we used different mutations than Qiao et al, we both analyzed the mutation R174E, which permits a direct comparison. This mutant retains ~50% ability to deaminate G4 DNA (Fig 5D in Qiao et al), yet has no CSR activity, as them and us show. We propose that the discrepancy between G4 deamination activity and CSR ability for AID R174E is due to the effect on licensing. The data supporting this interpretation is i) We show that the R-mutants are not only inactive for CSR but also for SHM at the IgV, which does not form G4 structure, and the new ChIP data show the same defect in occupying the gene body but not the TSS; ii) We provide new data showing that when we bypass licensing by using the constitutively nuclear AID Δ E5 variant, R174E or R178D do not prevent efficient mutation of the S μ (in fact AID Δ E5 R174E has normal activity compared to AID Δ E5), implying binding and deamination ability. Thus, at least for two of our mutants the lack of association of R-mutants with the gene body cannot be explained by the results of Qiao et al. Together with the interactions lost by BioID, these results support our transcription elongation coupling model. We do make the provision that in the case of R171Y, a biochemical defect may contribute to the defect, in line with the findings of Qiao et al, which are not ruled out by our results.

Reviewer #2 (AID, B VDJ/SHM)(Remarks to the Author):

In their manuscript entitled “A licensing step links AID to transcription elongation for B cell mutagenesis”, the authors mainly focused on three arginine residues in AID’s α -6 motif, and

found that these R-mutants mutate E.coli with the same efficiency as wildtype AID but fail to induce CSR and SHM (Fig 1). Then the authors made efforts to elucidate the underlying mechanism. On the one hand, they showed that the R-mutants have normal DNA binding kinetics (Fig 1), unchanged ssDNA deamination activity (Fig 2), normal nucleocytoplasmic shuttling (Fig 2), regular association with nuclear complexes (Fig 4), and similar dependence on Spt5/RNAPII for chromatin association (Fig 5). On the other hand, the authors showed that the R-mutants exhibit reduced association with the Igh Su locus (Fig 3 and 7) and with a series of proteins, including Spt6 (Fig 6). From these observations, the authors conclude that R-mutants lack association with the transcription elongation machinery and lack a licensing mechanism for AID's mutagenic activity.

Although the authors convincingly showed that these specific replacements of arginine residues abrogate SHM and CSR without affecting AID's enzymatic activity, overall the manuscript would benefit if attention were paid to the following.

We thank the reviewer for appreciating our structure-function work and for prompting a more careful consideration of our model, in particular regarding the implications of the findings by Qiao et al (Mol Cell 67, 361-73, 2017). We have added several new data that provide additional evidence of our model and discussed our findings compared to those of Qiao et al, which are not mutually exclusive. These changes have strengthened the manuscript and we hope are sufficient to ease the Reviewer's concerns.

Major points:

1) The observations described in this manuscript can be explained by recently published AID structure and in vitro data. As acknowledged by the authors in the Discussion part, it has been shown recently by AID structure why the mutations on RR domain affect AID deamination activity. In Qi's paper (Qi et al. Mol Cell. 67, 1-13; Fig 5), it is shown that one AID molecule interacts with two single-stranded DNA (ssDNA) overhangs: one is the substrate for deamination, while the other is for assistance to enhance affinity. This "assistant patch" is exactly the RR domain mentioned in the authors' manuscript. Moreover, Qi and his colleagues also showed that mutations on R171, R174, R177, and R178 specifically compromised AID deamination activity on substrates with two ssDNA overhangs, but not significantly altered on substrate with only one ssDNA overhang. These structure and in vitro data in Qi's paper perfectly explained the authors' observation in this manuscript that these R-mutants abolish activity for CSR but still maintain enzymatic activity. However, the authors' explanation seems less convincing and the novelty is limited.

We thank the Reviewer for prompting a more detailed analysis of this point. We acknowledge that our original data, focused on the analysis of the S μ region, left open the possibility that a biochemical defect for deaminating structured DNA, as described by Qiao et al, might explain the defect of our R-mutants. The caveats to that interpretation, which we did not discuss well in the original manuscript, were that a) Qiao et al show that the assistant patch enhances deamination of forked and especially G4 DNA, but is dispensable for deamination linear DNA, in vitro. Our biochemical assays were done using a bubble substrate, which is more similar to a forked DNA than to a linear substrate (Fig 1e, f) and show no defects for the R-mutants; b) The R-mutants were active to deaminate in E coli and were inactive for SHM at the IgV, which does not form structured DNA; c) The possibility that the R-rich region has a G4-independent role has already been noted in the commentary accompanying Qiao et al; based on the lack of G4 structures in SHM targets, as well as the fact that AID can efficiently trigger CSR from S regions that do not form structures (Pucella and Chaudhuri. Mol. Cell

67, 355–357 ; 2017). We have now included additional data to support our interpretation and improved our Discussion.

We added new data showing that the defect for SHM and off-target damage of the R-mutants correlates with their depletion at the IgV and Il4 α gene bodies (**new Figures 7c-g**). These data strongly suggest the results of Qiao et al do not explain the defects of the R-mutants. If they did, our data would imply that SHM and most off-targets susceptible to AID-induced DNA damage need to adopt structured DNA conformations to be deaminated in vivo, which to our knowledge has not been suggested.

We now show directly that in the context of the C-terminally deleted AID variant AID Δ E5, which bypasses licensing, mutating the RR domain does not prevent deamination of the IgV or, more critically, of the structured S μ region. The efficiency of each mutant is different but R174E and R178D produced a substantial number of mutations, similar to unmutated AID Δ E5. Moreover, both produce widespread DNA damage detectable as γ H2AX foci, which suggest they are active genome wide (**new Figures 8d-g**). These results show that our R-mutants do not prevent intrinsic ability to deaminate structured DNA at the S μ in vivo.

Finally, please note that our results do not rule out the existence of an assistant patch in AID involving some of these Arg residues. The impaired deamination activity of AID Δ E5 R171Y in B cells might be partly explained by the results of Qiao et al. However, we would like to respectfully note that the study of Qiao et al also has limitations. They could not co-crystallize AID with DNA and the position of the assistant patch was deduced from modeling of a truncated AID version that was fused to MBP. Which residues actually contact the DNA remains to be determined. While biochemistry with purified full length enzyme on oligonucleotides clearly shows that mutations in alpha-6 Arg reduce the activity on G4 DNA, the situation might be different in the context of transcription, which was not explored by Qiao et al. As we note in our discussion, Qiao et al and us used different mutations, but we both tested AID R174E for CSR. We both find a virtually complete CSR defect, yet this mutant retains substantial activity for deaminating structured oligonucleotides (at least 50%, estimating from Qiao et al Fig 5D). Thus, it is possible that part of the CSR defect displayed by the mutants tested by Qiao et al might be explained by the licensing defect we describe. We have briefly discussed these points in the modified Discussion.

2) The conclusions in the last part of the manuscript need some additional strengthening. The authors conclude that "AID R-mutants fail to progress with elongating RNAPII", and it is based on only two facts: 1) AID R-mutants interacts less abundantly with Spt6, which may participate in transcription elongation; and 2) AID ChIP qPCR data showed that the R-mutants were depleted from Su region but still enriched at the TSS. This conclusion needs to be further strengthened, for example, by experiments to show that AID loading is only reduced around Su region but not at TSS in Spt6-deficient cells. Moreover, genome-wide ChIP Sequencing needs to be performed so that other switch regions or off-target sites could be examined.

Our main finding is the existence of a licensing step for AID, which is in itself an important advance in understanding AID targeting. We propose that licensing operates by coupling AID to transcription elongation. In the revised manuscript we include new qChIP data to support our model, as requested. We do not have experience in ChIP seq, which we will pursue in the future. However, we have used qChIP to extend our analysis to critical AID target regions: the acceptor S-region and a well-known off-target gene of AID in B cells undergoing CSR to IgG1, as well as the IgV in cells

undergoing SHM. In every case, we see that the R-mutants can occupy the region around the TSS at similar levels than wt AID, yet they are always depleted in the gene body (**new Fig. 7**). Together with the demonstration that at least two of the R-mutants are able to substantially deaminate the S and IgV regions when licensing is bypassed, this result makes a compelling case to consider that the physical or functional coupling with transcription elongation is involved.

To comment on the evidence gleaned from the BioID experiments: We pursued Spt6 because it was a known AID interactor with a demonstrated role in CSR. The link of Spt6 with transcription elongation has been made by others (for example Andrulis et al. High-resolution localization of *Drosophila* Spt5 and Spt6 at heat shock genes in vivo: roles in promoter proximal pausing and transcription elongation. *Genes Dev* 14, 2635-49, 2000). But the BioID evidence suggesting that the R-mutants lose interaction with transcription elongation goes beyond Spt6. Other factors identified point in the same direction: notably CDK9 but also Nap114, another histone chaperone, as well as RNA processing factors. All these are novel AID interactions that deserve to be analyzed but this is beyond the scope of this work, as exploring each interaction is a project.

3) The proposed licensing mechanism lacks evidence. The authors concluded that the licensing steps could be bypassed by extending AID's residence in the nucleus (Fig 7d and 7e). However, as shown in Fig 3b, R-mutants did not totally abolish their deamination activity. Therefore, accumulated nuclear residence leading to high mutation level is not very surprising. Moreover, deletion of AID's C-terminal nuclear exporting signal would affect AID's stability. Therefore, the authors need to show that $\Delta E5$, $\Delta E5$ -R171Y, $\Delta E5$ -R174E, and $\Delta E5$ -R178D have similar protein stability in Fig 7d and 7e.

We thank the Reviewer for raising these points. We have added new data supporting the licensing mechanism, as detailed above. However, we respectfully disagree with the interpretation that the rescue of AID SHM activity by the constitutively nuclear AID $\Delta E5$ R-mutants variant is an expected result. All of the mutants show similar mutagenic activity in *E coli* (**Fig. 8b**). If increased nuclear residence were sufficient to homogenize their mutagenic potential in B cells, they should all result in equal levels of SHM or DNA damage. In the same vein, if the defect of the R-mutants was borne in a biochemical deficiency, increasing their nuclear concentration would be expected to increase SHM and DNA damage ability proportionally. However, this is not the case. We have produced new data showing that in the context of AID $\Delta E5$ each of the mutations permits deamination at the IgV or S μ at different levels (**Fig 8f, 8g, supplementary Fig. 6**) that is not proportional to their activity in *E coli*. While R174E has no effect on the activity at these regions in the context of AID $\Delta E5$, R178D only slightly reduces it and R171Y reduces it considerably. This reveals an additional defect of R171Y that we could not detect otherwise. This relative difference can be observed as well in their ability to produce off-target DNA damage (new data in **Fig 8d**). The reason for these differences were mentioned above and are discussed in the revised manuscript.

As pointed out by the Reviewer, the truncation of the C-terminal region of AID not only leads to full nuclear localization but also reduces the half-life of AID (Aoufouchi et al. *J. Exp. Med.* **205**, 1357–1368, 2008). As suggested, we checked the expression levels of the AID $\Delta E5$ R-mutants to rule out that the mutations might increase the stability of the proteins, resulting in higher nuclear levels. **New Fig 8h** shows that AID $\Delta E5$ and all three AID $\Delta E5$ R-mutants are equally expressed in B cells, at levels that are lower than the largely cytoplasmic full length AID, as expected.

In the Discussion part, the authors proposed "high nuclear AID levels can bypass the licensing step, possibly by direct recruitment to stalled polymerases." If this were true, deletion of AID's C-terminal nuclear would lead to higher interaction with Spt5, but the authors did not provide these data.

We agree that higher Spt5 association would be expected. We have attempted such experiments using GFP-tagged versions of AID and AIDΔE5 (because the available anti-AID antibodies for IP do not recognize AIDΔE5) to co-IP endogenous Spt5. Please note the large difference in protein expression between AID and AIDΔE5, caused by the differential stability of AID in cytoplasm and nucleus (Aoufouchi et al. J. Exp. Med. 205, 1357–1368, 2008). As shown in the figure below, we do see an increased association of AIDΔE5 with Spt5, once we normalize for relative protein levels. However, even if the result is as our model would predict, given the potential caveats posed by technical limitations we prefer to be cautious and not include it in the manuscript. The result would be available with the published revision files, as is customary for Nature communications, but the interested audience could read about our concerns.

Whole cell lysates from HEK293T cells expressing A2, AID or AIDΔE5 with C-terminal GFP fusion, were subjected to anti-GFP pull down. GFP and SPT5 were detected by WB from either immunoprecipitates or lysates. For each immunoprecipitate, SPT5 binding was measured as the SPT5 signal intensity relative to GFP signal intensity, and normalized to AID-GFP. One of three independent co-immunoprecipitations is shown, and quantification from all three plotted.

Additional points:

1) In the arginine mutation experiment, R174 is mutated to E; while R177 to A, and R178 to D. What is the reason for these different mutations? The authors stated, "AID R177A maintained all three activities". R is positively charged; D/E are negatively charged; and A is non-polar. Therefore, if the positively charged R177 were mutated to negatively charged D or E, maybe the CSR or SHM would be affected.

We apologize that the rationale for choosing the mutations was not clear. This is now explicitly mentioned in the in the first section of Results and we provide more details below.

The reason for our choice derives directly from our initial approach. Since we first identified α -helix 6 as a region of interest by using AID-APOBEC2 chimeras, once we started dissecting individual residues we chose to mutate each AID α -helix 6 residue to the corresponding APOBEC2 residue. The reason for this choice was to increase the chances of reproducing the phenotype with a single mutation, while hopefully preserving putative structural elements intact. This choice seems validated by the fact that changing R171 or R174 to lysine affects catalytic activity (Supplementary Fig. 2).

As mentioned by the Reviewer, it is possible that mutating R177 to D or E may have an effect (note that Qiao et al tested R177D only in combination with R171D or R178E, so the phenotype of R177D alone for deaminating structured DNA is unknown). As in any structure-function analysis by single amino acid replacements, the phenotype obtained is a combination of the identity of the residue replaced and the residue introduced. We mutated R177 to A, because that is the corresponding residue in A2. Our objective was to obtain catalytically active but functionally inactive AID variants, and so we continued with the three R-mutants that reproduced the phenotype. Nonetheless, we had previously mutated R178 to Ala and found that the effect on SHM and CSR was similar to that of R178D (included here below), demonstrating the phenotype of R178D does not simply reflect the change in charge. Though this does not remove the possibility that R177 is also involved in the licensing mechanism and/or the proposed assistant patch, we believe that there would be an effect of R177A if it was involved. In any case, we do not claim to have exhaustively characterized the residues involved in licensing AID, but the identification of licensing.

Left, Mutagenic activity in *E. coli*, measured by the relative frequency of rifampicin (Rif) resistant colonies arising from cultures expressing AID variants or empty vector (Ctrl). Means (bars) + SD obtained from 3 independent experiments (5 cultures/experiment) are shown, normalized to AID. **Middle**, somatic hypermutation activity, assayed by the relative IgM-loss accumulation in cultures of DT40 *Aicda*^{-/-} $\Delta\Psi\lambda$ B cells complemented with AID variants-ires-GFP or empty vector (Ctrl). Means (bars) + SD of the median values obtained from 3 independent experiments (≥ 12 cultures/experiment) were normalized to the median value of AID. **Right**, class switch recombination activity, monitored in naïve *Aicda*^{-/-} mouse primary B cells cultured with LPS and IL-4 and transduced with AID variants-ires-GFP or empty vector (Ctrl). The proportion of IgG1+ cells in the GFP+ population was determined 72 h after transduction. Means (bars) + SD of 3 independent experiments with 2 mice per experiment, are shown, normalized to AID.

2) It is a little confusing that in Fig 6g R-mutants co-immunoprecipitated Spt6, while in Fig 6h R-mutants could not interact with Spt6. The author reasoned it as "the power of BioID to detect functional defects in live cells". It would be appropriate to explain more here or to give a reference.

Our interpretation of this observation is rooted in the very different nature of each assay. BioID detects proteins only if and when they come in close proximity to the bait. Importantly, as BioID is done in live cells, it preserves compartmentalization and can sample dynamic interactions, while Co-IP does not have these abilities. On the other hand, Co-IP can detect interactions that are possible between proteins that come together after cell lysis even if they were not happening in the live cells in the conditions assayed. We interpret our results accordingly. AID and Spt6 interact only at a certain point of AID function, which we postulate is during transcription elongation. This interaction is lost by the R-mutants according to BioID, which is therefore detecting the loss of a functional interaction. However, BioID is insufficient to determine whether the ability to physically interact is also lost, as this would be indistinguishable from the proteins simply not coming close to each other in the

particular cellular conditions when the assay was performed. On the other hand, once cells are lysed for CoIP, the large pool of cytoplasmic AID becomes available to interact with the abundant Spt6 released from the nucleus. The assay essentially becomes a pull down that informs more about the ability of the proteins to interact biochemically than about the functional relevance of that interaction. In this case, the function of Spt6 and its functional link with AID are known from previous work from Dr Honjo's group, as referenced in the manuscript.

The fact that the R-mutants are able to interact with Spt6 by coIP but not in live cells, as probed by BioID is more informative than either assay alone. Our conclusion is thus based on both results taken together. Our data provides evidence of the uncoupling between the R-mutants and transcription elongation in live cells and suggest that coupling to Spt6 during transcription elongation is a feature of AID licensing. At the same time, as we would expect that mutations in the RR domain disrupt the physical interaction with the key licensing factor, our results also suggests that Spt6 is not the immediate licensing factor. We have reworded the Results section and our interpretation is now included in Discussion.

3) It is very interesting that a series of interaction proteins with AID were identified. A lot of them have not been reported before. Further investigation would facilitate the understanding of the functional deficiency of R-mutants.

We agree that these other factors are interesting and we will pursue them. However, each new interaction could be a self-standing study that we believe is better done in a separate study where we can fully describe their role.

4) Fig 3a and Fig 7a are repeated data, which could be combined.

5) In page 12, "Fig. 3f-h" needs to be changed to "Fig. 3e-g".

Thank you for these two suggestions. The changes have been made.

Reviewer #3 (AID, VDJ, RAG, DNA repair) (Remarks to the Author):

In this study, Methot and colleagues showed that three arginine residues in AID $\alpha 6$ domain are critical for both CSR and SHM in B cells, but are dispensable for cytidine deamination in E. coli or in vitro. The arginine mutations impair the interaction of AID with transcription elongation and correspondingly reduce the recruitment of AID to $\Sigma\mu$. Based on these data, the authors conclude that the arginine residues play an important role in coupling transcription elongation with AID function, and refer to the coupling mechanism as a "licensing step" in AID-mediated mutagenesis. It has been known that AID function is closely linked to transcription, but the underlying mechanism is not fully understood. This study represents a significant step forward towards solving this important question. Overall, the data on the characterization of AID mutants in various assays are convincing and the study is appropriate for Nature Communications. The authors could improve the study by addressing to the following comments:

We are grateful for the positive comments and thoughtful suggestions about our work. We have addressed all the Reviewer's concerns and implemented the suggestions.

1) A central piece of evidence for the licensing model is the observation that AID R-mutants occupy the promoter region, but not the body of $S\mu$. To generalize the model, the authors may wish to extend this analysis to an additional physiological target of AID such as another S regions or Ig variable region if the latter is experimentally feasible. In any case, it would be helpful if they could include a highly transcribed non-target gene for as a control for the specificity of AID ChIP.

We thank the reviewer for this suggestion. This concern was shared by all reviewers and we made a strong effort to address it. We present new ChIP data for the Sy1 and the AID off-target Il4r α in complemented *Aicda*^{-/-} primary B cells, as well as the IgV of DT40 cells undergoing SHM. We also included Gapdh, promoter and gene body as control for the ChIP specificity. All the new ChIP data is included in new **Fig. 7**. We refer to the response to reviewer 1, comment 6 for further details.

2) The authors argue that the licensing mechanism is enforced by limiting nuclear levels of AID. To strengthen this claim, the authors might use the ChIP assay to demonstrate that E5 deletion can rescue the recruitment defect of R-mutants.

We agree with the Reviewer and have strengthen this claim.

*Firstly, we have quantified the levels of nuclear AID achieved by full length AID versus AID Δ E5 in B cells (using GFP fusions because of the limitation to detect the truncated form), which shows that the latter leads to an average of 3-fold higher nuclear enzyme levels in steady state (**Fig. 8a**).*

*Secondly, we provide evidence that the E5 deletion rescues the recruitment defect caused by the R-mutants. Unfortunately, as our ChIP antibodies recognize the E5 region of AID, it is not possible to ChIP AID Δ E5. As a proxy to measure recruitment, we have performed mutation analysis by the AID Δ E5 R-mutants at the IgV and $S\mu$ regions (**Fig. 8f, 8g**). In addition, we show that AID Δ E5 R-mutants induce gamma-H2AX foci formation, as a proxy for recovered off-target targeting. As detailed in the answers to point 6 of Reviewer 1 and point 3 of Reviewer 2, AID Δ E5 R174E and AID Δ E5 R178D generated similar mutation frequency than AID Δ E5 at both regions and produced abundant off-target DNA damage. AID Δ E5 R171Y was still compromised for mutating the $S\mu$ and IgV and for DNA damage, which reveals that R171Y produces an additional defect. This fact is now acknowledged in the revised manuscript, but the presence and frequency of mutations at the $S\mu$ and IgV by the other two mutants demonstrate occupancy of the gene bodies, made possible by the higher nuclear levels of the AID Δ E5 form.*

3) In figure 2, the authors should consider complementing IF assay with Western analysis of AID protein levels in nuclear and cytoplasmic fractions.

*The WB is now included as **Figure 2b**.*

4) R-mutations decrease the interaction between AID and Spt6. However, the authors provided no evidence to demonstrate that the loss of this interaction is in fact responsible for the CSR or SHM defect. This point should be mentioned.

The Reviewer is correct and we thank the notice. We do not demonstrate that the loss of interaction in live cells between the R-mutants and Spt6 causes the licencing defect. We used this observation, and the loss of other novel interactions of AID that we report in this paper, to infer that the R-mutants might have a defect to couple or link with transcription elongation. We precise this point in the Discussion.

5) As shown in Figure 1c and d, the expression level of R174E protein is lower than WT protein. Furthermore, as judged by visual inspection of the image in Figure 2a, R174E protein remains largely cytoplasmic even in the presence of LMB and DidB. Thus, the authors should note the possibility that the CSR and SHM defect associated with R174E mutation may be attributed to reduced expression and nuclear accumulation.

It is true that total AID R174E is less expressed than WT. However, we do not believe this to explain its defects because WB and quantitation of nuclear and cytoplasmic extracts (Figure 2b, 4d, e) show that AID R174E shows reduced cytoplasmic levels but the same nuclear abundance wt AID. This is probably the result of increased nuclear shuttling. We note that R174E is more responsive to LMB (see Fig 2a), which according to our previous work (Methot et al J. Exp Med 212, 581-596; 2015) reflects reduced cytoplasmic retention and/or increase nuclear import.

REVIEWERS' COMMENTS:

Reviewer #1 (Remarks to the Author):

This is an excellent manuscript, and has been significantly improved. It advances the field significantly and is also provocative in the ideas and models raised. Many groups will now be thinking about the mechanism of licensing; this might be a key step forward for field, which has been struggling for many years to understand what enables AID to act where it does in the genome.

I have only one comment. The new AID ChIP data of Figure 7 are something of a technical tour de force, as many groups have struggled (unsuccessfully) to detect AID reliably by ChIP at sites outside of S regions. This said, the ChIP signals, while apparently above background, are EXTREMELY small. For example, the data in the gene body of IL4Ra relies on a difference between pulling down roughly 4 alleles out of 10,000 for WT AID and 2 alleles out of 10,000 for the R mutants. These numbers might very well reflect what actually happens in cells, and I'm certainly not suggesting that any more experiments be done (e.g., I don't think AID ChIP-seq would add anything here given the extremely low signals produced). But I would be more comfortable with the text relating to Figure 7 (pages 13-14) if some acknowledgement were made of the low signal, for example, by noting that ChIP signals were low but reproducible. Relying on a 2 fold difference at what must be the very limit of the sensitivity of ChIP (with its well known propensity for variability and artifact) is a risky proposition.

Reviewer #2 (Remarks to the Author):

The revision is excellent and the paper should now be accepted

Reviewer #3 (Remarks to the Author):

The authors have addressed all of our comments in a Satisfactory manner. We recommend that this very nice paper should now be accepted for publication in Nature Communications.

REVIEWERS' COMMENTS (on the revised manuscript):

Reviewer #1 (Remarks to the Author):

This is an excellent manuscript, and has been significantly improved. It advances the field significantly and is also provocative in the ideas and models raised. Many groups will now be thinking about the mechanism of licensing; this might be a key step forward for field, which has been struggling for many years to understand what enables AID to act where it does in the genome.

I have only one comment. The new AID ChIP data of Figure 7 are something of a technical tour de force, as many groups have struggled (unsuccessfully) to detect AID reliably by ChIP at sites outside of S regions. This said, the ChIP signals, while apparently above background, are EXTREMELY small. For example, the data in the gene body of IL4Ra relies on a difference between pulling down roughly 4 alleles out of 10,000 for WT AID and 2 alleles out of 10,000 for the R mutants. These numbers might very well reflect what actually happens in cells, and I'm certainly not suggesting that any more experiments be done (e.g., I don't think AID ChIP-seq would add anything here given the extremely low signals produced). But I would be more comfortable with the text relating to Figure 7 (pages 13-14) if some acknowledgement were made of the low signal, for example, by noting that ChIP signals were low but reproducible. Relying on a 2 fold difference at what must be the very limit of the sensitivity of ChIP (with its well known propensity for variability and artifact) is a risky proposition.

We thank the Reviewer for appreciating the work and revision. We are aware of the difficulties in detecting AID by ChIP outside the S-regions. We have adapted our methods accordingly, by using stringent controls and calibration curves for the qPCR. Please note that we subtract the IgG controls from the signals. Also, please note that overexpression caused by the retroviral complementation of the AID-deficient cells probably help us in detecting the signal, compared to other work attempting to detect endogenous AID. Nonetheless, the signal at the Il4ra is low, as noted by the Reviewer. However, our use of stringent background and negative controls, together with the consistency of the results, make us confident that we are detecting AID and that we can reproducibly distinguish 2-fold differences, as shown. The same applies to the results at the IgV in DT40.

Reviewer #2 (Remarks to the Author):

The revision is excellent and the paper should now be accepted

We thank the Reviewer for appreciating our efforts to revise the manuscript.

Reviewer #3 (Remarks to the Author):

The authors have addressed all of our comments in a Satisfactory manner. We recommend that this very nice paper should now be accepted for publication in Nature Communications.

We thank the Reviewer for the positive comments.